# Subjective nature of path information in quantum mechanics

Xinhe Jiang [1,2,5], Armin Hochrainer[1,2,5], Jaroslav Kysela [1,2], Manuel Erhard[1,2], Xuemei Gu [1,3], Ya Yu[1,4] & Anton Zeilinger [1,2] ✉

Common sense suggests that a particle must have a definite origin if its full path information is available. In quantum mechanics, the knowledge of path information is captured through the well-established duality relation between path distinguishability and interference visibility. If visibility is zero, high path distinguishability can be achieved, which enables one to determine with high predictive power where the particle originates. We investigate the complementarity between path information and interference visibility through an experiment involving three sources emitting into identical modes. Our findings challenge the classical intuition that a particle can be traced back to its origin through its trajectory when full path information is available. By grouping the crystals in different ways, we demonstrate that it is impossible to ascribe a definite physical origin to the photon pair, even if the emission probability of one individual source is zero and full path information is available. Our results shed new light on the physical interpretation of probability assignment and path information beyond its mathematical meaning and show that the interpretation of path information in quantum mechanics is subjective.

Bohr's complementarity principle[1] is a cornerstone in quantum mechanics, illustrating the mutual exclusivity of certain properties of a system—such as wave-like interference and particle-like path information. The particle nature is usually characterised by the ability to distinguish between the two paths and to analyse which slit the particle passed through, as famously exemplified by the double-slit interference experiment. Physically, the complementarity principle manifests itself as a question of predictability: could one consistently win with more than 50% probability by betting on the outcome of a which-path measurement? High path distinguishability enables such predictive power but eliminates interference visibility. Conversely, if interference fringes are observed with high visibility, no meaningful path information can be obtained, and path alternatives become entirely unascertainable. Quantitatively, the trade-off between path *distinguishability* (*D*) (which-path information) and *visibility* (*V*) of the interference fringes is encapsulated in the duality relation $D^2 + V^2 \leq 1$[2–5],

indicating that observation of an interference pattern and acquisition of which-path information are mutually exclusive. This duality relation has been extensively studied in two-path interferometers[6,7].

A striking example of illustrating the duality relation is frustrated down-conversion[8], an interference phenomenon that occurs when photons are generated from two sources[9]. In this setup, photon pairs may be emitted from either crystal, and the two possibilities are aligned so that they overlap perfectly in both spatial and temporal modes. If the two sources are indistinguishable, which means that spontaneous parametric down-conversion (SPDC) photons from each source have the same spatiotemporal modes, same frequency and bandwidth, etc., perfect interference can occur, i.e. the interference visibility can reach 1. However, if one source is blocked or misaligned, the photons' origin becomes distinguishable, and interference is lost. Frustrated down-conversion induces the idea of path identity, which has led to many striking quantum interference effects and quantum

[1]Institute for Quantum Optics and Quantum Information, Austrian Academy of Sciences, Vienna, Austria. [2]Vienna Center for Quantum Science and Technology, Faculty of Physics, University of Vienna, Vienna, Austria. [3]Max Plank Institute for the Science of Light, Erlangen, Germany. [4]Shanghai Jiao Tong University, Shanghai, China. [5]These authors contributed equally: Xinhe Jiang, Armin Hochrainer. ✉e-mail: anton.zeilinger@univie.ac.at

applications in recent years[10,11]. Path distinguishability, in the context of frustrated down conversion, means information about which source produced the photon pair. If interference is reduced, complementarity allows for a more profitable bet on the outcome of a 'which-source' measurement due to the increased amount of information concerning the source of the photons. Frustrated down-conversion thus provides a good platform to study the interplay between interference and which-source information.

The duality relation has also been extended to multi-path interference[12–16]. In such cases, interference between multiple alternative ways of emitting photon pairs can be observed, and these observations have consistently confirmed the duality relation between visibility and distinguishability[17,18]. In this work, we used a frustrated SPDC system with three nonlinear crystals to explore a different aspect of the duality relation, namely, the interpretation of path information when applying the theoretical formalism to an actual experiment. This room for interpretation stems from the different ways of partitioning reality into alternatives, represented by probability amplitudes. We find that interpreting distinguishability as information about the origin of photon pairs is inconsistent in a three-crystal setup. Unlike the two-crystal case, ascribing a definite origin to a photon pair is impossible, even though full 'path information' is available and no interference is observed if the two crystals are grouped together. This result arises because, while the quantum-mechanical formalism provides an unambiguous and objective description of the system, the interpretational narrative used to describe the underlying processes can be subjective.

Here, we show that there are multiple valid ways to assign probability amplitudes to the alternatives in the experiment, leading to incompatible interpretations of the results. In general, a given probability can be decomposed into alternative probability amplitudes, each of which may be non-zero, while their coherent superposition leads to a vanishing total amplitude. This implies that 'which-path information' can only be meaningfully defined if the context is clarified and the relevant alternatives are explicitly specified in this context. This undermines the view that path information is a definitive and objective property of the system.

## Results
### Inconsistent which-path information in three-crystal interference

Considering that three nonlinear crystals are pumped with the same laser, which emits photon pairs into identical modes (Fig. 1a), the quantum state of the system can be written as[8,19],

$$|\psi\rangle = ae^{i\phi_A}|s\rangle|i\rangle + b|s\rangle|i\rangle + ce^{i\phi_C}|s\rangle|i\rangle \tag{1}$$

where $ae^{i\phi_A}$, $b$, $ce^{i\phi_C}$ represent the probability amplitudes of photon pair creation at the respective crystal, $|s\rangle|i\rangle$ represents a signal and idler photon pair in the modes that can be detected, and $\phi_A$, $\phi_C$ are the relative phase between the photons emitted from each pair of crystals. This state is consistent with the Hamiltonian-based models of the SU(1,1) interferometer[20], which are fundamentally different from those of the traditional Mach–Zehnder interferometer. In Supplementary Note 1, we show that our setup is actually a nonlinear interferometer with three processes. With this, the rate of the emitted photon pairs can be written as

$$R(\phi_A, \phi_C) \propto |ae^{i\phi_A} + b + ce^{i\phi_C}|^2 \tag{2}$$

As stated, the amount of 'which-source' information and visibility exhibit a trade-off relation due to the complementarity principle. By treating two of the three sources as a single source and applying the duality relation (the applicability of the duality relation to our setup is given in Supplementary Note 2), we show that analysing the 'which-

source' information and visibility from this point of view leads to a contradiction between two possible interpretations of the same experiment.

First, consider the first two nonlinear crystals (NL1 and NL2) to constitute one source labelled by S1, while NL3 constitutes a second source S2, as shown in Fig. 1b. With this treatment, the quantum state of the photon pair in Eq. (1) can be rewritten as

$$|\psi\rangle = [\alpha + ce^{i\phi_C}]|s\rangle|i\rangle \tag{3}$$

where $\alpha = ae^{i\phi_A} + b$ is the probability amplitude corresponding to photon pair emission by S1 and $ce^{i\phi_C}$ corresponds to an emission by S2. Thus, for a fixed relative phase $\phi_A$ between NL1 and NL2, defining the parameter $\alpha$, the rate of emitted photons can be rewritten as

$$R(\phi_C) \propto |\alpha + ce^{i\phi_C}|^2 \tag{4}$$

Now, one can apply the duality relation between the observed visibility upon varying $\phi_C$ and the amount of available 'which-source information'. Consider an experimental situation in which $a = b$ and $\phi_A = \pi$. It follows that $\alpha = 0$. Therefore, the probability amplitude corresponding to a pair of photons emitted by S1 is equal to zero. Thus, S1 is 'switched off'. This is consistent with the observation that S1 corresponds to a frustrated down-conversion tuned to completely destructive interference. Consequently, if S2 is removed, in principle, no pair of photons is emitted from the system. However, if instead the first 'double' source (S1) is blocked or removed, the photon pair emission rate is unchanged. Therefore, there will be zero interference visibility by varying the relative phase $\phi_C$ between S1 and S2. These observations are consistent with the duality relation $D^2 + V^2 \leq 1$, as the sources S1 and S2 are unbalanced and the information from 'which-source' is available here. One might conclude that all photon pairs come from source S2, that is, from NL3. However, this interpretation leads to a contradiction, as is shown below.

Instead of grouping events into photon-pair emissions from S1 and S2, consider an alternative view of the experiment. Crystal NL1 is considered the first source S1′ and the combination of NL2 and NL3 is considered the second source S2′, as indicated in Fig. 1c. The possible events are identified accordingly as photon pair emission from either S1′ or S2′. In this situation, the quantum state is

$$|\psi\rangle = [ae^{i\phi_A} + \beta]|s\rangle|i\rangle \tag{5}$$

with the probability amplitude for photon pair emission by S1′ being $ae^{i\phi_A}$ and the amplitude for S2′ is $\beta = b + ce^{i\phi_C}$. For a fixed relative phase $\phi_C$ between NL2 and NL3, defining the parameter $\beta$, the rate of emitted photons can be rewritten as

$$R(\phi_A) \propto |ae^{i\phi_A} + \beta|^2 \tag{6}$$

Similarly, we can make the following argument. Taking into account the situation where $b = c$ and $\phi_C = \pi$, it follows that $\beta = 0$. Again, the duality relation $D^2 + V^2 \leq 1$ can be applied in a self-consistent way. Individual blocking of S1′ and S2′ shows that S2′ does not emit photon pairs. In this case, S2′ is 'switched off'. No interference visibility will be observed when the phase $\phi_A$ is varying. This corresponds to full 'which-source' information. Thus, one can conclude that the photons were emitted by S1′, that is, NL1.

It should be noted that both of these two situations can be realised in one experiment, allowing $a = b = c$ and $\phi_A = \phi_C = \pi$. With this setting, the corresponding rate of detected photons is non-zero and the same for these two situations through Eq. (2). If we fix one phase to be $\pi$ and scan another phase, this rate will remain constant, and there will be no interference visibility. At the same time, we still have no information about which crystal produced these photons. We can see that both

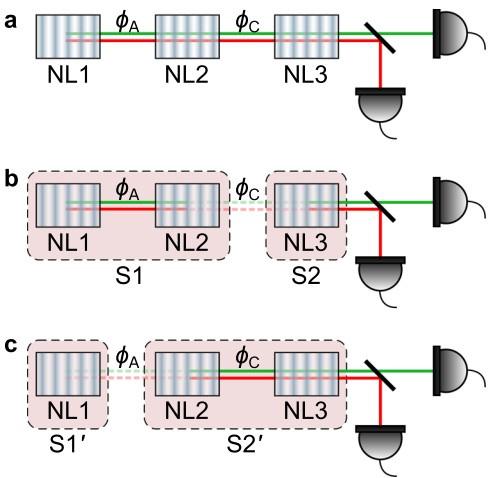

**Fig. 1 | Schematic of three-crystal interference. a** The three crystals emit photon pairs in identical modes. **b** The combination of NL1 and NL2 is considered a single photon pair source (S1). The second source (S2) consists only of NL3. The relative phase $\phi_A$ between NL1 and NL2 changes the probability of S1 emission. If set to $\pi$, the photon pair emission of S1 is completely suppressed. Thus, all emitted photons come from NL3 and no visibility of the total emission rate is observed when varying the relative phase $\phi_C$ between S1 and S2. **c** NL1 corresponds to the source S1′, while the combination of NL2 and NL3 is considered a single source S2′. With the same argument, we conclude that all the photons must have originated from NL1 and no visibility is observed when varying the relative phase $\phi_A$ if $\phi_C = \pi$. These two contradictory situations can be realised in the same experiment when the intensities are balanced and $\phi_A = \phi_C = \pi$. Note that the phase shifter introduces a negative relative phase between NL2 and NL3. Due to its symmetry and periodicity, we represent it as $\phi_C$.

viewpoints are equally self-consistent and obey the duality relation. However, the interpretations provided by analysing the 'which-source' information in two alternative ways of partitioning the system into two separate sources are incompatible with each other.

## Experimental demonstration of inconsistent which-path information

The experimental setup is shown in Fig. 2. A 405 nm pump light is used to pump the three periodically poled potassium titanyl phosphate (PPKTP) crystals. The 810 nm photon pairs (signal and idler) are generated in a collinear arrangement through a SPDC process with a type-II configuration. The three crystals emit photons into identical modes. After the third crystal, the pump light is filtered with a dichroic mirror (DM) and a band-pass filter (BPF). Finally, the SPDC photons are detected with single-photon detectors and sent to the coincidence logic for processing. The pump power is low enough so that only one photon pair at a time is present. Phase plates are placed between each pair of crystals and are used to tune the relative phase between the pump, signal, and idler photons, and thus changing the relative phase $\phi_A$ and $\phi_C$. The lenses between each pair of crystals form a 4f system to obtain good spatial overlap of the SPDC photons from each crystal.

To obtain good interference visibility between all crystals, the spectral and spatial degrees of freedom of the SPDC photons should be indistinguishable. This can be realised with identical crystals and by overlapping their spatial mode through fine-tuning the position of lenses in the setup. To facilitate this, we used an intensified charge-coupled device (ICCD) camera to image the SPDC photons. The lens system allows us to successively image signal and idler beams originating from each of the SPDC processes at the crystal and at its Fourier plane. The photon beams from the three SPDC processes are aligned to overlap in both planes to ensure that they are indistinguishable in any plane. The details and characteristics of the imaging are shown in the

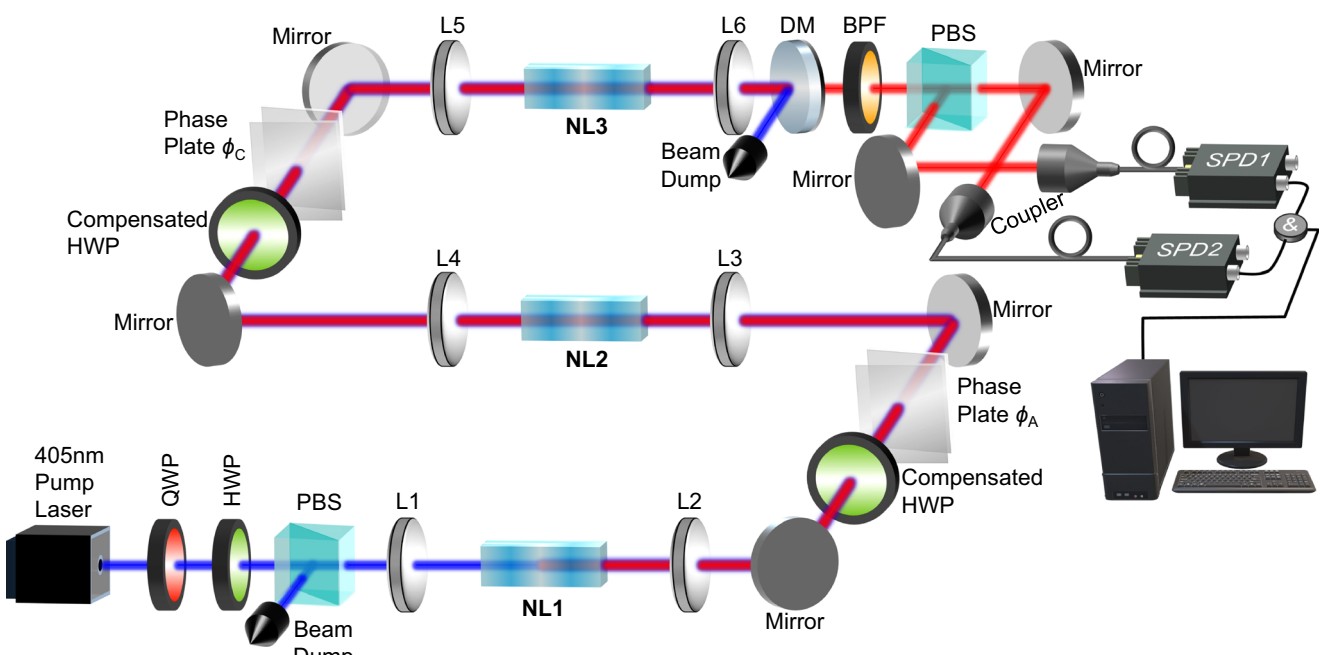

**Fig. 2 | Experimental setup of the three-crystal interference.** The 405 nm pump laser is used to pump the three crystals. The SPDC photons are generated and emitted in identical modes. Each crystal is enclosed by two lenses, and the two lenses between each pair of crystals form a 4f system. This ensures that the first crystal is imaged onto the second crystal and a good spatial overlap of the beams created in the two SPDC processes is obtained. The phase plates between each pair of crystals are used to change the relative phases ($\phi_A$ and $\phi_C$) of the pump, signal, and idler photons. To compensate for the longitudinal walk-off caused by the different group velocities of signal and idler photons, we used two compensated half-wave plates between each pair of crystals. The photons are finally filtered and collected in the fibre for coincidence counts. QWP quarter-wave plate. HWP half-wave plate. PBS polarising beam splitter. NL1-NL3 non-linear crystals 1-3. L1-L6 lenses 1-6. DM dichroic mirror. BPF band-pass filter. SPD single-photon detector. &: coincidence logic.

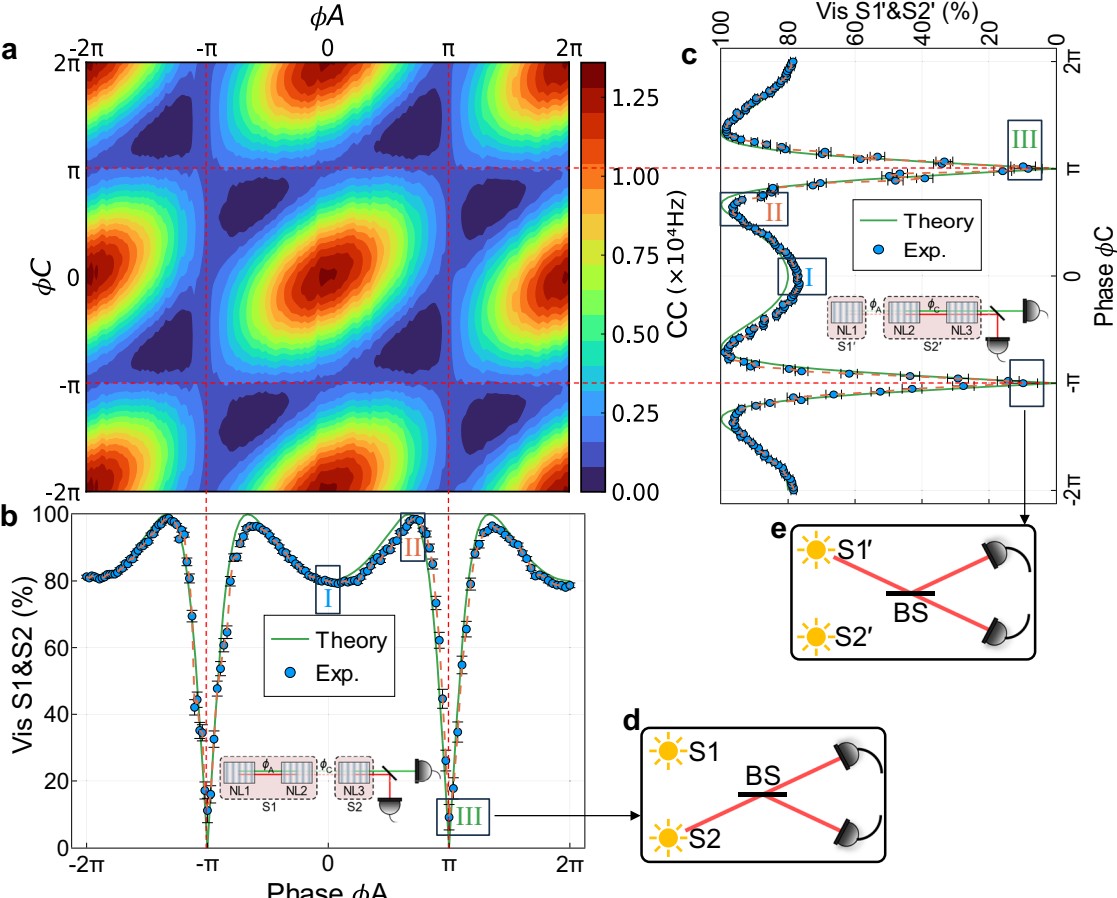

**Fig. 3 | Three-crystal interference obtained with a 2D scan. a** Contour plot shows the detected coincidence counts (*CC*) when all three crystals are active upon varying both phases $\phi_A$ between NL1 and NL2, and $\phi_C$ between NL2 and NL3. **b** Visibility observed between S1 and S2 when each phase $\phi_A$ is fixed and phase $\phi_C$ is scanned. The inset shows the first perspective, which regards the system as a two-process interferometer between S1 and S2. At the phase point $\phi_A = (2k+1)\pi$ ($k\in \mathbb{Z}$), the photon pair emission from S1 is suppressed, as shown schematically in **d**. **c** Visibility observed between S1′ and S2′ when each phase $\phi_C$ is fixed and phase $\phi_A$

is scanned. The inset shows another perspective, which regards the system as a two-process interferometer between S1′ and S2′. In an analogous manner, the photon pair emission from S2′ is suppressed at the phase point $\phi_C = (2k+1)\pi$ ($k\in \mathbb{Z}$), as shown schematically in **e**. The coincidence count data in frames I, II, III of **b**, **c** are shown in Fig. 4a, b, respectively. The green lines in **b** and **c** are the theoretical prediction. The blue dot is the experimental data. The dashed orange line is a guide for the eye. BS beam splitter.

'Methods', 'Characterisation of SPDC photons'. In addition, the crystal temperature is also optimised to ensure a degenerate photon pair emission from all three crystals.

To get a full view of how visibility changes when one phase is fixed and the other phase is scanned, we set different values for one phase and obtain visibility by scanning the other phase, as shown in Fig. 3. As it is periodic, only some periods are given. Figure 3a shows the contour plot when all three crystals are active, and the two phases are varied. For comparison, the theoretical contour plot is given in the Supplementary Fig. 3b. First, phase $\phi_A$ is set to different values, and the visibility is calculated when phase $\phi_C$ is scanned (see Fig. 3b). As can be seen, visibility is highest when the phase is set to $\phi_A = 2\pi/3$. This is because the amplitude of the combination of NL1 and NL2 is more balanced compared to the amplitude of NL3 at this point. They have better interference visibility as a result of their better indistinguishability. However, at the point $\phi_A = \pi$, they show almost no interference due to complete distinguishability. Similar results are also obtained when the phase $\phi_C$ is set to different values and the visibility is measured by scanning the relative phase $\phi_A$ (see Fig. 3c). The inset of Fig. 3b, c shows the two pictures when the crystals are grouped differently. At the phase point $(2k+1)\pi$ ($k\in \mathbb{Z}$), the photon pair emission is suppressed, corresponding to one of the sources (S1 or S2′) being switched off in the two-source interferometer, as shown schematically

in Fig. 3d, e. Therefore, the knowledge that photons can only be emitted by the other source (S2 and S1′) yields full interferometric path information in the sense that one could always win a bet on the outcome of the measurement on the individual sources.

In Fig. 4a, b, we plot the interference fringe at three representative phases 0, $2\pi/3$ and $\pi$, which are labelled I, II and III in Fig. 3b, c. Phase 0 has higher counts but lower visibility. Phase $2\pi/3$ has lower counts, but the highest visibility. Phase $\pi$ exhibits almost no visibility. More theoretical analysis and comparison with experimental data can be found in 'Methods'. To take a closer look at these two pictures, one of the phases is fixed as $\pi$, and the other phase is scanned. According to the theoretical analysis, we will see a constant count rate. In our experiment, the visibility of the interference between the probability amplitude contributed by S1 and that of S2 when setting $\phi_A = \pi$ is 9.12 ± 3.81%. The corresponding visibility between S1′ and S2′ when setting $\phi_C = \pi$ is 8.30 ± 4.19%. This is almost close to the random noise level when there is no interference at all. In Fig. 4c, d, we plot the coincidence counts (*CC*) with error bars when $\phi_A = \pi$ and $\phi_C = \pi$, respectively. It can be seen that *CC* fluctuates within a small range. This even reaches the random noise level. In addition, the two cases have overlapping count regions (1600–1800 Hz). In this region, we cannot tell at all from which crystal the photons come from. Using the counts data, we estimate that the path information corresponds to photon

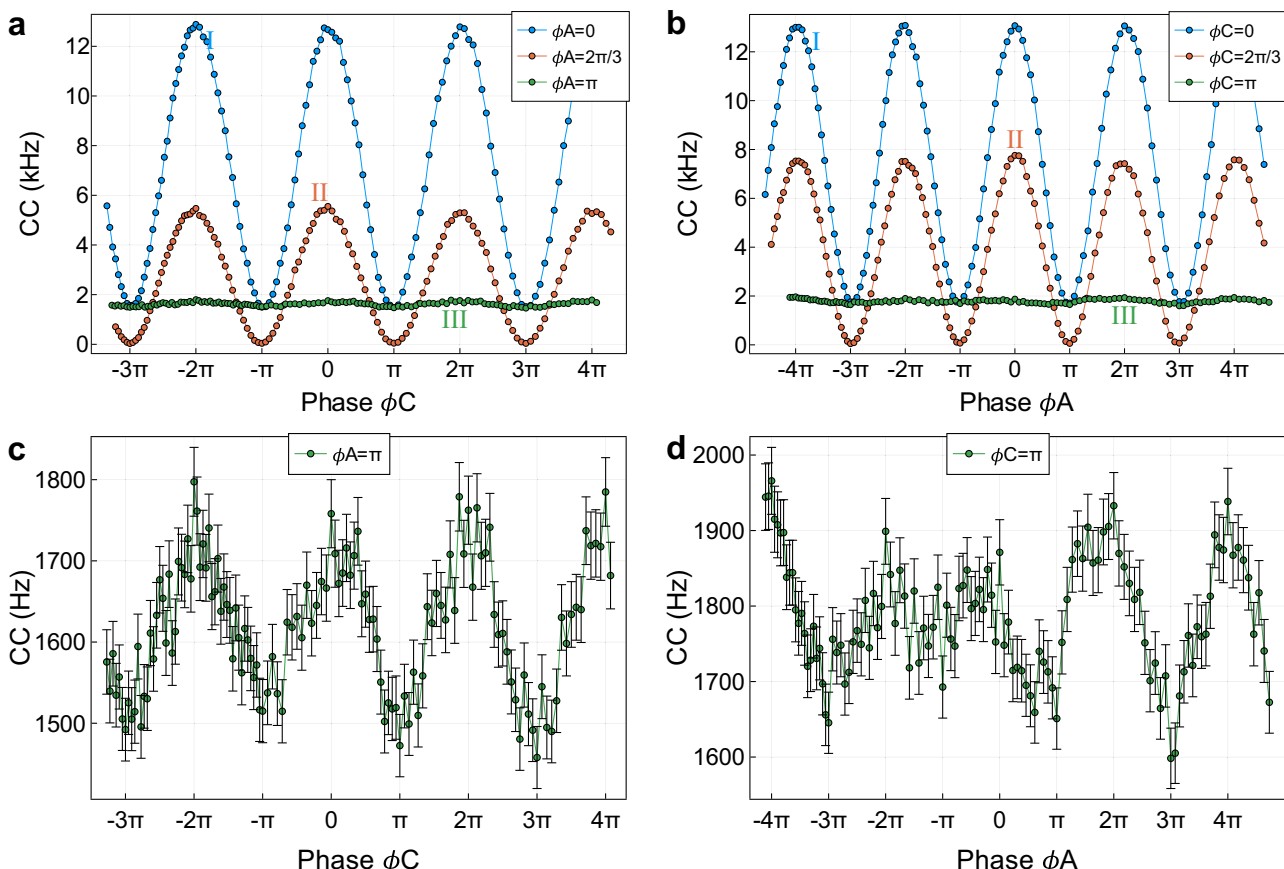

**Fig. 4 | Interference fringes at the phase points 0, 2π/3, and π. a, b** Coincidence counts (*CC*) for the three points I: $\phi_A = 0$ ($\phi_C = 0$), II: $\phi_A = 2\pi/3$ ($\phi_C = 2\pi/3$), III: $\phi_A = \pi$ ($\phi_C = \pi$) chosen in Fig. 3b, c, respectively. **c** *CC* versus $\phi_C$ when relative phase $\phi_A = \pi$. **d** *CC* versus $\phi_A$ when relative phase $\phi_C = \pi$. The lines are guides to the eye. Errors are determined by assuming Poissonian counting statistics.

pairs originating from NL3 with the probability $p_3 = 95.14 \pm 0.59\%$ when $\phi_A = \pi$ and from NL1 with the probability $p_1 = 96.41 \pm 0.47\%$ when $\phi_C = \pi$ (probability $p_1$ is calculated by $CC_{NL1}/CC_{tot}$ and probability $p_3$ is calculated by $CC_{NL3}/CC_{tot}$, where $CC_{NL1} = CC_{tot} - CC_{S2'}$ and $CC_{NL3} = CC_{tot} - CC_{S1}$, $CC_{S1}$ and $CC_{S2'}$ are counts when the crystals NL3 and NL1 are off, respectively, and the corresponding phase $\phi_A$ and $\phi_C$ is set to $\pi$, $CC_{tot}$ are the counts when all three crystals are on and both phases $\phi_A$ and $\phi_C$ are set to $\pi$). Using these data, the duality relation obtained in our experiment is shown in the Supplementary Table 1. As $p_1 + p_3 > 1$, we arrive at a contradiction because it is impossible to have a probability larger than 1 in an experiment carried out with $\phi_A = \phi_C = \pi$. It should be noted that it is not possible to have equal overlapping regions and a constant count due to experimental imperfections. Therefore, visibility cannot completely become zero at phase $\pi$. The analysis of some of the experimental errors is shown in 'Methods', 'Optimisation of the visibility'.

## Discussion

Note that in a periodically poled crystal, the amplitudes of pair creations at different crystal sections are engineered to interfere constructively to increase the pair-creation efficiency. Usually, a single probability amplitude is assigned to photon pair emission within the whole crystal, but no description is given anymore at which position inside the crystal the photon originates. In an analogous fashion, we can also group two crystals together and assign a single probability amplitude for the photon-pair emission. However, the assignment of a probability amplitude of zero to an alternative should not be interpreted as a zero probability that this alternative will occur. In the

experiment of two-crystal frustrated down-conversion[8], if one crystal emits no photons, one can determine with high probability that the detected photons originate from the second crystal. Nevertheless, the probability must include all the contributions of the three crystals if a third nonlinear crystal were inserted before the detectors. Different from the two-crystal case, a measured probability of zero for the first two crystals does not mean that no photons come from them in the presence of a third crystal, since each crystal contributes to a probability amplitude alternative, which finally interferes with each other[19].

Another interesting point is to realise the subtle difference in the operational and physical interpretation of the 'which-path information' depending on what measurement is performed[19]. If the three crystals are analysed separately, 'which-path information' refers to the question 'which crystal generates the photons before they arrived at the detectors'. If two of them are grouped together and assigned one common probability amplitude, the 'which-path information' then corresponds to the question 'which of the two possible events happened? Did the photons come from the first crystal or from a combination of the other two?' Despite these different perspectives, the amount of path information remains entirely dependent on visibility via the duality inequality. Therefore, in the situation of both phases set to $\pi$, all these pictures are equally valid and self-consistent, although they lead to incompatible which-path information.

Bohr's complementarity principle is usually understood as the wave-particle duality relation (WPDR). The visibility of an interference pattern is used to quantify the wave property, and the path information is used to quantify the particle property. The one-particle and two-particle interference are widely characterised in the framework of the

duality relation. This interference phenomenon has its root in the indistinguishability of the different alternatives, which contribute to the overall quantum amplitudes of the system as a whole. In single and two-particle interferometry, this complementarity is explored through the relationship between path distinguishability and the visibility of interference fringes[2,4,21–23]. Compared with two-particle interference, there exist entirely different implications in three-particle interferometry that are worth further investigation[24]. Our experiment also demonstrates an important feature of quantum mechanics: the description of the interference of a system of quantum emitters (even if it is composed of spatially distinct parts in a laboratory) must encompass all possible alternatives according to the sum of the corresponding amplitudes, as long as no intervention is made to make these alternatives distinguishable. Whether the three-crystal system should be treated as a whole or can be analysed separately depends on whether their contributions to these different alternatives can be distinguished or not. This point of view may renovate our understanding of the interplay between whole and part in the context of distinguishability.

We note that some theoretical works have provided a general framework to derive WPDRs based on entropic uncertainty relations (EURs)[25]. The equivalence of WPDR and EUR has also been demonstrated in a recent experiment for two-path interferometers[26]. In the entropic view, visibility and distinguishability can be defined through some kind of entropy, which quantifies the information of the particle and wave behaviour obtained in the system. The extended framework for formulating the WPDR from EUR in multi-path interferometers[27] provides experimental insights into investigating the WPDR and its equivalence to EUR in higher dimensions, based on our current setup. This may be the subject of our follow-up work.

In this work, we showed that both no which-path information and no interference could be realised in a single experiment by involving three indistinguishable sources. By treating two of the three sources as a single source, two incompatible interpretations of the path information are possible within a single experimental configuration. This brings about a refinement of the physical interpretation and experimental description of 'which-path information'. Usually, we identify macroscopically distinguishable 'alternatives' and assign probability amplitudes to them. However, this identification of 'alternatives' is generally ambiguous. Our findings suggest that path information depends not only on the physical system, but also on how to translate the formula to an actual experiment. This offers a fresh perspective on quantum interference. By revealing the subtleties of frustrated downconversion in multi-crystal setups, this work deepens our understanding of quantum phenomena and raises important questions about the nature of path information in quantum mechanics.

## Methods
### Theoretical analysis
In this section, we present more theoretical results of the three-crystal interference. As stated in the main text, the rate of the emitted photon pairs can be calculated through the Eq. (2).

Assuming that the three crystals emit photons with the same amplitude, that is, $a = b = c = 1$, the count rate can be obtained when varying the two phases. The count rate versus the phases $\phi_A$ and $\phi_C$ is shown in Supplementary Fig. 3. Note that at the point $\phi_A=\phi_C=\pi$, there is a non-zero photon pair emission rate with zero visibility.

The cross-section for different phases $\phi_A$ is shown in the Supplementary Fig. 4. Due to symmetry, the result is the same for phase $\phi_C$. We choose several representative points to see how the visibility changes with the phases. We can see that the visibility is highest at the point $2\pi/3$ and becomes zero at the point $\pi$. The counts of $\pi/3$ ($2\pi/3$) and $5\pi/3$ ($4\pi/3$) are symmetric with respect to $\phi = \pi$. Therefore, they show the same visibility. To compare them, the minimum (maximum) point of the experimental data is aligned at $\pi$ ($2\pi/3$) in the main text

(Fig. 3). Due to periodicity, the two points $\phi_A = 0$ and $\phi_A = 2\pi$ overlap each other. One feature is that the minimal and maximal counts at the middle phase point (e.g. $\phi_A = \pi/3$) are higher than at the lowest visibility point (i.e. $\phi_A = \pi$). To provide a comprehensive comparison, we again show the visibility over the period from $-2\pi$ to $2\pi$ in Supplementary Fig. 5. Theoretically, the visibility is 0 at phase $\pi$ and 1 at phase $2\pi/3$. The visibility is 80% when the relative phase is 0.

### Characterization of SPDC photons
To guarantee that the SPDC photons from the three crystals have a good spatial overlap, we placed crystals one by one and imaged the SPDC photons with an ICCD camera. The setup of the imaging system is shown in the Supplementary Fig. 6. This system allows for successive images of both SPDC processes at the crystal and at its Fourier plane. Without L7 in place, the lenses L2, L3, and L6 map the Fourier plane of the crystals to the camera plane. If the lens L7 is inserted, the two lenses after each crystal make up an imaging system that produces an image of the crystal in the camera. Therefore, the photons from the SPDC processes of each crystal are aligned to overlap in both planes, which ensures their indistinguishability in any plane. After this alignment, the lens system and camera are removed from the setup using a flip mirror (FM). The images of the SPDC photons from the three crystals are shown in Supplementary Fig. 7. We can see that they are almost the same size at both the Fourier plane and the crystal plane after good alignment.

The interference visibility between each pair of crystals is first optimised in the experiment. The results are shown in Supplementary Fig. 8. The visibility is quantified by Vis = $(CC_{Max} − CC_{Min})/(CC_{Max} + CC_{Min})$, where $CC$ is the coincidence count rate. After recursive optimisation, the observed visibility between NL1 and NL2 is $V_{12}$=98.53 ± 0.18% and the visibility between NL2 and NL3 is $V_{23}$=98.68 ± 0.17%. For the source pair NL1&NL2 and NL2&NL3, the count rates can be expressed as in the non-ideal interference,

$$R_{12}(\phi) = A^2 + B^2 + 2AB \cdot \cos e1 \cdot \cos \phi$$
$$R_{23}(\phi) = B^2 + C^2 + 2BC \cdot \cos e3 \cdot \cos \phi \qquad (7)$$

$A$, $B$ and $C$ are amplitudes corresponding to each crystal, $e1$ and $e3$ accounts for the degree of coherence in the non-ideal case. Defining $s1 = A(\cos e1|1, 0, 0\rangle + \sin e1|0, 1, 0\rangle)$, $s2 = B|1, 0, 0\rangle$, $s3 = C(\cos e3|1, 0, 0\rangle + \sin e3|0, 0, 1\rangle)$ in the three-mode Fock space $|1, 0, 0\rangle$, $|0, 1, 0\rangle$, $|0, 0, 1\rangle$, the quantum state can be constructed as,

$$|\psi\rangle = s1 + s2 \cdot e^{i\phi_1} + s3 \cdot e^{i(\phi_1 + \phi_2)} \qquad (8)$$

Then, using the number operator $\hat{n} = \hat{a}^\dagger \hat{a}$, one can estimate the visibility between NL1 and NL3 without NL2, i.e. $s2$=0, and assuming $\phi_1$=0 without loss of generality. Based on the measured counts and visibility $V_{12}$ and $V_{23}$, the estimated visibility of NL1 and NL3 is $V_{13}$=97.24 ± 0.25%. This indicates that each pair of crystals exhibits a high degree of coherence.

Note that the spectral and spatial distributions of the SPDC photons from the PPKTP crystal are correlated. Both are influenced by the pump wavelength, the bandwidth of the BPF, and the crystal temperature. To graphically illustrate the phenomena, we schematically plot the angle distribution of the SPDC photons versus the wavelength at different crystal temperatures, as shown in Supplementary Fig. 9a. The filter for different bandwidths is also plotted. One observation is that the SPDC photons appear to be located in a ring when the temperature is low. It then becomes a spot as the temperature increases. The narrower the bandwidth of the filter, the higher the temperature it needs for the photons at the centre to appear. The image measured with the temperature change is shown in Supplementary Fig. 9b–g. These results are consistent with previous work[28]. Usually, we need to adjust the temperature at which the SPDC photons at 0° just appear

within the bandwidth, for example, the curve of temperature $T_3$ in Supplementary Fig. 9a. This can ensure that the photons have a good collinear overlap. The intersection point of the H and V photons is the degenerate point, which is at the double of the pump wavelength. In the experiment, it is hard to adjust the central wavelength of the BPF and the degenerate point to be the same. This will cause asymmetry of the H and V photons. When the H- and V-photons exchange the roles with the compensated HWP between each pair of crystals, the photons from the two crystals will have some distinguishability because of this asymmetry. This is one of the sources of errors that cause visibility degradation.

### Comparison of experimental and theoretical data

In the main text, we compare the experimental and theoretical visibility for the 2D scan. Here, we calculate the theoretical counts when one phase is fixed, and the other phase is scanned, and compare it with our experimental data. For simplicity, we pick up three representative points, that is, 0, $2\pi/3$, and $\pi$. The counts versus the phase are shown in the Supplementary Fig. 10. We can see that the visibility at phase $2\pi/3$ is the highest, i.e. 1 in theory. The visibility at phase $\pi$ is 0 in theory. The visibility at phase 0 is 80% in theory. Our experimental data are in good agreement with the prediction of the theory.

### Optimisation of the visibility

In the experiment, the SPDC photons must have a good spatial and spectral overlap to have a good indistinguishability. The spatial overlap can be easily realised by imaging the photons and adjusting the lens system and the translation and tilt of the crystals. To have a good spectral overlap, we use a BPF. Some unwanted photons, which cause the distinguishability, can also be filtered with a single-mode fibre. In addition, the crystal temperature also needs to be optimised back and forth. To ensure coherent emission of these crystals, the optical path-length difference (OPLD) should also meet some requirements. One is that the OPLD between the pump beam and the two down-converted photons must be smaller than the coherence length of the pump laser[10],

$$|(L_{P_i} + L_{SPDC_i}) - (L_{P_j} + L_{SPDC_j})| \overset{i \neq j}{\underset{(i,j) \in \{1,2,3\}}{\leq}} L_P^{coh} \qquad (9)$$

where $L_{P_{i/j}}$ are the optical path length of the pump beam from the pump laser to each crystal, $L_{SPDC_{i/j}}$ represent the optical path length of the respective down-converted photons from each crystal to the detector, and $L_P^{coh}$ denotes the coherence length of the pump laser. Another condition is that the OPLD of the down-converted photons should be less than their coherence length,

$$|(L_{S_j} - L_{I_j}) - (L_{S_i} - L_{I_i})| \overset{i \neq j}{\underset{(i,j) \in \{1,2,3\}}{\leq}} L_{SPDC}^{coh} \qquad (10)$$

with $L_{S_{i/j}}$ representing the optical path length of the signal photons from each crystal to the detector and $L_{I_{i/j}}$ for idler photons, $L_{SPDC}^{coh}$ denoting the coherence length of the down-converted photons. The first condition is easy to fulfill. To fulfill the second condition, we placed a compensated half-wave plate in the path. This wave plate is used to exchange the polarisation of the signal and idler photons and to compensate for the longitudinal walk-off between them. This will make their OPLD within the coherence length of the down-converted photons.

The general procedure of optimisation is first to adjust the coupling to get the maximal coincidence counts from the first crystal. The second crystal then opens. The SPDC efficiency is adjusted by tilting and moving the position of the crystal. The purpose is to make the counts of the two crystals as equal as possible. Then the phase between the two crystals is scanned. The temperature of the crystals is tuned to maximise their interference visibility. In addition, the relative position

of the two crystals is tuned to further improve visibility. The third crystal is opened after a good visibility for the first two crystals is obtained. Meanwhile, the first crystal is closed. Then, the position and temperature of the third crystal are optimised to maximise the visibility of the last two crystals. To keep the visibility of the first two crystals, the second crystal is not touched. After good visibility is obtained, we return to the first two crystals. The second crystal is adjusted to ensure that both pairs have good visibility. This procedure is repeated many times until we get good visibility on both sides. The phase is very sensitive to air fluctuations and mechanical vibration. To stabilise the phase, we isolate the setup with a box. After this, we can do a 2D scan in the long run.

One thing is that visibility degrades over time. This is caused by some residual strain on the mount stages of each optical component. Note that any small disturbance can induce a change in the spatial position of the SPDC beams and thus lead to the distinguishability of the SPDC photons from each crystal. Based on the theoretical model, all the errors that degrade visibility can be ascribed to three cases, namely the longitudinal misalignment, the transversal misalignment, and the tilt of each optical component. We have an estimation of these errors using relation $V^2 + D^2 = 1$ and define the quantity $D$ as any non-overlap portion of the beam which leads to the distinguishability. The visibility degradation caused by these errors is shown in Supplementary Fig. 11. The transversal and tilt errors have the most important influence on the visibility. Because the beam size is focused at about 50 µm, any transversal deviation at this scale will cause a great reduction in visibility. The tilt error will be magnified with the propagation distance. As shown in Supplementary Fig. 11a, the two beams overlap at position 1 and deviate from each other with a tilt angle $\theta$. After distance $L$, they are fully distinguishable. With a realistic propagation distance of 400 mm between the two crystals, we get the variation of visibility versus the tilt angle. For our setup, the beams propagate longer distances before they are collected in the fibre. A 0.1° tilt will reduce visibility to 97%. Therefore, we must carefully optimise the position and tilt of each optical component and crystal.

Another factor that influences visibility is the photon yield imbalance. This is because the pump power in each crystal cannot be the same due to the non-unity transmission of the optical components between the crystals. Assume that the photon yields of NL1, NL2, and NL3 are $A$, $B$, and $C$, respectively. The ideal case is $B/A = 1$ and $C/A = 1$. We theoretically analyse the photon yield imbalance error, which is shown in Supplementary Fig. 12. We can see that the error has a significant influence on the phase points of $\pi$ and 0. With an imbalance of 0.9, the visibility at phase $\pi$ has already deviated from the ideal value of 0 to 10%. Based on our measurements, the photon yields are approximately $A = 2200$ Hz, $B = 2000$ Hz, and $C = 1800$ Hz for each crystal. By substituting these experimental values, our visibility shows good agreement with the theoretical results taking these realistic errors into account.

## Data availability
The data that support the findings of this study are available within the main text and its Supplementary Information. Source data are provided with this paper.

## Code availability
The code used for data analysis is available from the corresponding authors upon reasonable request.

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

## Acknowledgements

We thank R. Kindler for many helpful discussions. This work was sup-
ported by the Austrian Academy of Sciences, the European Research
Council [SIQS, Grant 600645 EU-FP7-ICT], the Austrian Science Fund
(FWF) [FoQuS, grant-DOI 10.55776/F40 and CoQuS, Grant-DOI
10.55776/W1210], and the University of Vienna [Quantum Experiments
on Space Scale, QUESS]. The authors have applied a CC BY public
copyright license to any author accepted manuscript version arising
from this submission.

## Author contributions

X.J., A.H., J.K. and M.E. carried out the experiment. X.J. collected the data
with assistance from J.K., X.G. and Y.Y. X.J. and A.H. analysed the data
and performed the numerical calculations. X.J. wrote the paper with the
input of all other authors. A.H. and A.Z. conceived the project. A.Z.
supervised the research. All authors discussed the experimental results.

## Competing interests

The authors declare no competing interests.

## Additional information

**Supplementary information** The online version contains
supplementary material available at

Anton Zeilinger.

**Peer review information** *Nature Communications* thanks Girish Kulkarni,
Tyler Volkoff and the other, anonymous, reviewer(s) for their contribu-
tion to the peer review of this work. A peer review file is available.

