## [Transparent Peer Review file · Nature Communications]

Subjective nature of path information in quantum mechanics

Corresponding Author: Dr Xinhe Jiang

Version 0:

Reviewer comments:

Reviewer #1

(Remarks to the Author)

Reviewer #2

(Remarks to the Author)

The authors show that in a system of three SPDC processes configured to emit indistinguishable photon pairs, a standard and natural notion of "which path" information is untenable. In particular, at a working point which minimizes interferometric visibility (and therefore maximizes "which path" information) there are two contradictory ways to ascribe the "which path" information to downconversion sources. I could answer "yes" to all the desiderata for publication, so I recommend that that the paper eventually be accepted. Just please address the comments below in a minor revision. I waive my right to anonymity in the peer-review process for this manuscript. (Tyler Volkoff, Los Alamos.)

Comments:

1. The abstract reads like a philosophy paper: "this perception of path information is problematic...", "...reshape our understanding of the whole and part in the context of distinguishability." Just use some clear terminology and state your result.

2. "Path distinguishability, in this sense, means the information of which source produced the photon pair."

"Path distinguishability, in the context of frustrated downconversion, means information about which source produced the photon pair."

3. It is crucial to define what is meant by "indistinguishable sources" in the sentence "If the two sources are indistinguishable...". You have already defined path indistinguishability, so define also what you mean by "indistinguishable sources" (e.g., downconversion processes having the same pump, same nonlinearity, etc.). I know that this is discussed in more detail later in the paper, but a note should be made here.

Also, what is the meaning of "perfect interference" in this sentence? I think it means, according to things you already defined, the possibility of $V=1$. If this is what you mean, then please state it.

4. "If there is no interference, it is possible to bet on the outcome of a "which-source" measurement that we know with a higher probability of where the photons come from and vice versa."

"If interference is reduced, complementarity allows for a more profitable bet on the outcome of a "which-source" measurement due to the increased amount of information

concerning the source of the photons."

5. "This stems from the different possibilities of partitioning reality into events..."

"This room for interpretation stems from the different possibilities of partitioning reality into events..."

6. "As stated, the amount of "which-source" information and visibility is exclusive due to the complementarity principle."

"As stated, the amount of "which-source" information and visibility exhibit a trade-off relation due to the complementarity principle."

7. Choose one term and be consistent throughout: "which-source" information or "source information"

Choose one term and be consistent throughout: "which-way" or "which path"

8. For the authors consideration: the perturbative description of the state in (1) is correct, and in agreement with the non-perturbative Hamiltonian dynamics of SU(1,1) networks with indistinguishable or partially distinguishable modes (T.J.V. "Relative phase and dynamical phase sensing in a Hamiltonian model of the optical SU(1,1) interferometer"). A non-perturbative description of SU(1,1) networks based on continuous-variable quantum circuits (e.g., Yurke/Klauder in the simplest case of two downconverters) would give results for the visibility that are inconsistent with predictions from the Hamiltonian dynamics at some order in the nonlinearity.

9. "Thus, the rate of emitted photons is given by..."

"Thus, for a fixed relative phase ϕ_A between NL1 and NL2 defining the parameter α , the rate of emitted photons can be rewritten as..."

10. Above (6),

"The rate of emitted photons is given by..."

"For a fixed relative phase ϕ_C between NL2 and NL3 defining the parameter β , the rate of emitted photons can be rewritten as..."

11. In the Discussion, I liked the analogy with a single periodically poled crystal-- regardless of what is happening to the pump phase as it propagates through the crystal, we just keep track of the total downconversion amplitude, not the individual amplitudes from the segments.

12. "...the description of the quantum system must encompass all the possible outcomes, as long as..."

"...the description of a system of quantum emitters (even if it is composed of spatially distinct parts in a laboratory) must encompass all the possible emission outcomes according to the sum of corresponding amplitudes, as long as..."

13. The optical path length difference analyses in the Methods section are important considerations for indistinguishability of the emitted pairs from the three crystals. The authors clearly understand this.

14. The theory of uncertainty relations and wave-particle duality has progressed to much more descriptive and nuanced versions of the $D^2 + V^2 \leq 1$ relation, even for binary interferometers (see, e.g., Coles et al. "Equivalence of wave-particle duality to entropic uncertainty"). Therefore, it is hard to be satisfied by the interpretation of the "which path" information simply in terms of a definite downconversion source-- maybe there is a tradeoff between the visibility in this experiment and a variance or entropy of some measurement that can be considered as a more appropriate notion of "which path" information in the system and which does not lead to any interpretive contradictions. Can the authors provide any insight?

Reviewer #3

(Remarks to the Author)

In this manuscript, the authors consider a frustrated SPDC system with three nonlinear crystals to investigate deducible "source information" when grouping the crystals in different ways. Theoretically, they have observed that: If (1) each crystal emits photon pairs of the same amplitude and (2) the relative phase between two sequential crystals in the photon's path is set at π , the overall photon-pair generation rate of any two sequential crystals combined vanishes. Since we can meet these two conditions simultaneously for both the first & second crystals, as well as the second & third crystals, this appears to create a paradoxical situation where, even though the overall generation rate is nonzero, the photon pairs are seemingly not generated from any pair of the crystals. They use this final observation to conclude that "it is impossible to ascribe a definite physical origin of the photon pair even if the emission probability of one individual source is zero and full path information is available". They have also performed carefully designed experiments to confirm their theoretical findings.

While their observations and findings are thought-provoking, some gaps leading to their conclusion need further clarification. To start with, the whole reasoning seems to rest on the following premise in the abstract: “if visibility is zero, a high path distinguishability can be obtained”. However, if we look at the visibility-distinguishability tradeoff given in [4], we see that these quantities satisfy an inequality, rather than an equality. This means that, even in that context, the above premise does not necessarily hold. Although the tradeoff inequality can be saturated for certain pure states, to our knowledge, it has not been established that the inequality holds as an equality for all pure states. Even if the equality does hold for all pure states in an interferometric setup, it is unclear whether this quantitative expression relating visibility and which-way information can be translated into a similar tradeoff between visibility and “which-source” information in the non-interferometric setup considered in this work. Importantly, in the former scenario, the amplitudes and the relative phase are between the two different arms/ kets, which is not the case here. To this end, it is also unfortunate that notions like “path information” and “source information”, which are central to the whole argument, were not properly introduced. Consequently, one cannot even readily verify the validity of claims like “full path information is available”; or are we supposed to understand this to mean that the visibility is zero?

Coming back to the observations, if we think of NL1 & NL2 as one source, the authors argued that their combined probability amplitude is zero. While this statement looks innocent, does this mean we are entitled to think of

“combined probability amplitude of A and B is zero = the *emission* probability of both A and B as zero”?

From Equation (1), there seems to be no legitimate reason to consider only the probability from summing the amplitude of two of the three terms, all associated with the *same* ket vector $|s\rangle|j\rangle$. Even if the above statement is valid, when $A = NL1$ & $B = NL2$, one must conclude that NL3 must have contributed to generating the photon pairs in this case. Similarly, if $A = NL2$ & $B = NL3$, one must conclude that the photon pairs are generated by NL1. This seems to be the basis for the authors’ claim that we now have two “contradictory which-way information”, when both $A = NL1$ & $B = NL2$, and $A = NL2$ & $B = NL3$ conditions are satisfied. However, at least three arguments tell us that such reasoning might not be proper, or must be trodden on carefully.

First, if we push this view further and think that interaction between the fields of the signal/ idlers from NL1 & NL2 happens before the interaction with NL3—after all, the laser field first passes NL1 before NL2, and finally NL3—then one may argue that the NL1 & NL2 combined emission probability amplitude must be zero first, while the non-zero probability emerges later after the laser hits NL. So, whatever is detected must be emitted from NL3. Then, visibility is zero, source knowledge = NL3, and so there’s no ambiguity of “which-source” knowledge; everything appears consistent.

Second, what prevents the possibility that *some* of the photon pairs are from NL1 & NL2, and some from NL2 & NL3? This point of view is consistent with the experimental results as well.

Third, recall Feynman’s important lesson about quantum theory: we have to add all the relevant amplitudes before computing the probability. So, it may simply be incorrect to think of zero emission probability of both NL1 & NL2 (or NL2 & NL3), as this possibility happens, via Equation (1), in superposition with the emission from NL3 (or NL1).

All these “inconsistencies” point to another potential pitfall in concluding from quantum mechanical predictions using classical “intuitions”. Presumably, this is the basis for the following passage in the abstract “our findings shed new light on the physical interpretation of probability assignment and path information beyond its mathematical meaning and reshape our understanding of the whole and part in the context of distinguishability.” However, is it not also evident from the first paragraph of the Discussion and conclusion that the lesson to be learned here is, to some extent, already known in [8] (published more than 30 years ago) and [19] (in the PhD thesis of one of authors, published more than 4 years ago). If so, in what way has this shed new light on our understanding?

To summarize, while we agree that this work is thought-provoking and can potentially help us appreciate better the role of interference in a “which-source” setup, its current presentation does not make a strong case for publication in Nature Communications. Besides the issues raised above, the authors can also help improve the clarity of the work by addressing the following issues:

Lines 323/343: The authors should explain clearly what constitutes a 4f system. The descriptions around Line 323 and those around Line 343 are not consistent. Is it defined by lenses between each pair of crystals (as described in the former), or is it a pair of crystals (as described in the latter)?

Lines 399 & 401: “canned” should have been “scanned”

Line 417, 741-742, 745, 762: The hyperlinks for Extended Data Fig. 1b, Extended Data Fig. 1, Extended Data Fig. 2, and Extended Data Fig. 3 do not link the Figures under Extended Data, but to the main text.

Lines 497-498: From the paragraph after Eq. (4), it seems like the off (on) status corresponds to one setting the amplitude to be equal in magnitude and the relative phase to π (or not). However, from the description given here, the phases are both set to π . Then, what distinguishes the on/ off status of the crystals NL1 or NL3?

Lines 485-505: Following the last comment, the coincidence count rates $CC_{\{NL3\}}$ and $CC_{\{NL1\}}$ were presumably carried out under different experimental settings (with different crystals turned off). If so, what entitles us to compare and even combine the probability of photon pairs generated in the individual crystals? And why should one expect that this has

anything to do with the situation when all crystals are turned on?

Line 1048: Did the authors mean to write $\phi_A = \pi$ or $\phi_C = \pi$?

Line 1072: How are the counts of Extended Data Fig. 2 normalized for different choices of ϕ_A ? And why are the curves for $\phi_A=0$ and $\phi_A=2\pi$ different?

Line 1214: It would be helpful to indicate either on the images or in the caption which images are for the crystal plane and which others are for the Fourier plane.

Reviewer #4

(Remarks to the Author)

Version 1:

Reviewer comments:

Reviewer #1

(Remarks to the Author)

please find attached

Reviewer #2

(Remarks to the Author)

The paper can be published now. I'm happy to have alerted the authors to the apparently open problem of developing entropic uncertainty relations which satisfactorily describe the tradeoff between visibility and an appropriate notion of distinguishability for compound nonlinear interferometers.

Reviewer #4

(Remarks to the Author)

We (reviewer 3) acknowledge the great effort by the authors in responding to the comments in our previous report.

Previously, the main point of the paper, and hence its value, was kind of fuzzy and not clear to me. In particular, I think the modification in the abstract, as well as elaboration in the conclusion, makes it now clearer. To my understanding, the main point of this paper is to bring up (or in some sense, to remind, because I think actually this is not completely unknown to the physicists, but usually we don't really care about this because we don't worry about interpretation) the point that, as the authors say, "Any probability could, in principle, be subdivided into alternatives of non-zero probability amplitude, which can lead to a zero-probability amplitude of the combined probability. Therefore, "which-path information" can only be meaningfully defined if the context is clarified and the alternatives are explicitly specified in this context" (The earlier version using "event" was somehow making it confusing). In fact, I think this sentence in the conclusion perhaps should be put earlier in the introduction, as this is really the central message in my opinion (which in fact goes beyond the discussion about nature of path itself, which now serves as a specific demonstration).

For the other points that the authors replied, and all the additions in the Supplementary material, as well in the main text, and corrections of typos, we are mostly OK with that.

Now, as for whether the work is suitable for publication in Nature Communications, to be honest, we remain somewhat reserved. As we said in our earlier report, we value that "this work is thought-provoking and can potentially help us appreciate better", now instead of "the role of interference in a "which-source" setup", I would say "the identification of 'alternatives' versus which-path information", still, I'm not certain if this work really brings something that is really fundamental for new physics in the future. So, I'm not completely against its publication in Nat. Commun., but I'm not completely recommending it either.

Peer Review File

Manuscript Title: Subjective nature of path information in quantum mechanics

REVIEWER COMMENTS & AUTHOR RESPONSES

Reviewer Reports on the Initial Version:

Reviewer #1 (Remarks to the Author):

I have carefully read the manuscript titled "*Subjective nature of path information in quantum mechanics*". In this work, the authors describe a quantum optical interference experiment in which the which-path question appears to admit two distinct but equally self-consistent answers. In what follows, I outline my queries and comments regarding this work:

- There appears to be a contradiction between

sentence (line 38) "*We demonstrate the simultaneous observation of zero interference visibility and the complete absence of which-path information using a three-crystal interference setup.*"

and

sentence (line 123) "*Unlike the two-crystal case, ascribing a definite origin to a photon pair is impossible even though full "path information" is available and no interference is observed.*"

The authors may please correct this discrepancy.

- On line 72, the word "outcomes" may be replaced by "path histories". Also, "unpredictable" does not seem fully accurate because this is a statement about retrodiction and not prediction. Perhaps "unascertainable" could be a better choice.
- On line 100, the sentence reads "*Frustrated down-conversion thus provides a platform to study the interplay between indistinguishability, interference, and which-source information.*" Perhaps the word "interference" may be redundant here because the tradeoff is between indistinguishability and which-path information.
- On line 118, the sentence reads "*This stems from the different possibilities of partitioning reality into events that are represented by probability amplitudes.*" As the term "event" refers to something that happens at a given location in space and instant in time, it does not seem appropriate here. I think the word "events" may be replaced by "path histories" for accuracy.
- On line 223, "*Now, we can apply the duality relation between the observed visibility*" may be rephrased to "*Now, one can apply the duality relation between the observed visibility*". Similarly, in line 278 "*Thus, we can conclude that the photons were emitted by S1', that is, NL1.*" may be rephrased to "*Thus, one can conclude that the photons were emitted by S1', that is, NL1.*"
- On line 317, the sentence reads "*Phase plates are placed between each pair of crystals and are used to tune the relative phase (ϕ_A and ϕ_C) between the pump, signal, and idler photons.*" In my view, the phrase "*between the pump, signal, and idler photons*" is vague. The quantities ϕ_A and ϕ_C are relative phases between probability amplitudes corresponding to the path alternatives of the process from the source to the detector.
- In the caption for Fig 3, there are multiple typos. Please correct them.
- On line 455, the sentence reads "*In our experiment, the visibilities of the interference between S1 and*

S2 and between S1' and S2' when setting $\phi_A = \pi$ and $\phi_C = \pi$ are $9.12 \pm 3.81\%$ and $8.30 \pm 4.19\%$, respectively." Here S1, S2, S1', S2' refer to effective sources. Strictly speaking, the interference is not between sources but rather between probability amplitudes corresponding to downconversion happening at those sources. Although I agree that brevity and simplicity may warrant some flexibility, the authors may want to explore other rephrasings. Also, do these visibilities follow any duality relation or tradeoff with respect to the path probabilities p_3 and p_1 calculated on line 492-294? If yes, I would suggest that the authors point this out.

- On line 493, the sentence in the parentheses "(the probability is calculated by $CCNL1 / CC_{tot}$ and $CCNL3 / CC_{tot}$, where $CCNL1 = CC_{tot} - CCS2'$ and $CCNL3 = CC_{tot} - CCS1$, $CCS1 / CCS2'$ are counts when crystal NL3/NL1 is off and the phases are set to π , CC_{tot} are the counts when all three crystals are on and the phases are set to π)" is confusing. It is not clear which quantity refers to p_1 and which refers to p_3 . Please specify explicitly. Also, when the authors say "the phases are set to π ", they are probably referring to ϕ_A and ϕ_C , but it is better to specify as "both phases ϕ_A and ϕ_C are set to π ".
- On line 524, the authors write "However, the assignment of a probability amplitude of zero to an event should not be interpreted as a zero probability that this event will occur." As I have pointed out before, the use of the word "event" is problematic. The authors are referring to an entire "path history" or "path alternative" from the pump source to the detector. The probability amplitudes correspond to these alternatives, and not to specific events.
- On line 526, the authors write "In the experiment of frustrated down-conversion [8], ideally no photon pairs are detected if there is complete destructive interference between two photon-pair emission processes. Nevertheless, the possibility for a photon pair originating from frustrated down-conversion cannot be ruled out if a third nonlinear crystal were inserted before the detectors. Therefore, a measured probability of zero for an isolated subsystem does not mean that this probability remains zero in the presence of other subsystems [19], since all subsystems can interfere with each other."

I disagree with the above reasoning. When an additional crystal is added to the system, the system has changed in a non-trivial manner. There is no reason to expect that the photon pair rate would remain zero. Mathematically speaking, the total probability amplitude for the entire process must now an additional term. Contrary to what the authors say, the sum of the two original terms still remains zero. It is the introduction of the new term corresponding to the third crystal that will now make the total photon pair rate non-zero. There is nothing confusing or surprising about this. The situations with and without the third crystal are two completely different situations.

Also, it is not accurate to say "subsystems can interfere with each other" because interference does not happen between systems (or subsystems) per say. This issue was at the heart of Dirac's remark that a photon only interferes with itself, and not with any other photon. Roy Glauber, for instance, later clarified that interference happens between probability amplitudes corresponding to path histories or path alternatives. And these alternatives are specified all the way from the original source to the detection stage. This is the basic premise of even the path integral formalism pioneered by Feynman.

- On line 561, the authors write "The one-particle and two-particle interference are widely characterized in the framework of the duality relation. This interference phenomenon has its roots in the fundamental indistinguishability of identical quantum particles."

I disagree with this remark. The origin of interference is not the indistinguishability or the identicalness of the particles themselves, but rather of the alternatives or the path histories of the system as a whole. That system could be a one-particle system or a two-particle system depending on the kind of experiment being performed.

Note that if what the authors state was true, it should never be possible to observe two-photon interference effects with non-degenerate entangled photons from parametric down-conversion. However, Hong-Ou-Mandel-type effects and other two-photon interference effects have been observed with two-photon systems comprising of non-identical photons. In fact, the famous postponed

compensation experiment reported in the paper titled “Can two-photon interference be considered the interference of two photons?” *Phys. Rev. Lett.* 77, 1917 (1996) shows a Hong- Ou-Mandel-type effect with the photons not even appearing at the beam-splitter at the same time. Such experiments clearly demonstrate that it is not the indistinguishability or identicalness of the photons themselves that counts, but rather of the alternatives of the evolution of the system as a whole.

- On line 573, the authors write “Our experiment also demonstrates one important feature of quantum mechanics: the description of the quantum system must encompass all the possible outcomes, as long as no intervention is made to make the parts distinguishable in principle. Whether the quantum system should be treated as a whole or as separable parts depends on whether the parts can be distinguished from each other.”

I don't quite agree with this claim. Strictly speaking, the experiment has only two possible outcomes, namely, either the photon pair is detected or it is not. But the description must encompass all the possible alternatives or path histories, not outcomes. In addition, it is not really a question of making the parts distinguishable, but rather about making the alternatives distinguishable. The quantum system under investigation is the two-photon field, not the crystals. And it is imperative to treat this two-photon system as a whole right from the pump source to the detectors. The present study does not raise any questions about this matter.

- On line 586, the authors write “By considering two of the three sources as a single source, two contradictory interpretations of the path information are possible within the experimental configuration.” I am not sure “contradictory” is the right word here. Within the quantum formalism, there is no really no contradiction. There are two distinct common-sense interpretations that are equally self-consistent but mutually incompatible from a common-sense point of view.
- On line 591, the authors write “Usually, we identify macroscopically distinguishable “events” and assign probability amplitudes to them. However, this identification of “events” is generally ambiguous. Any event could in principle be subdivided into subevents of non-zero probability amplitude, which can lead to a zero probability amplitude of the combined event. Therefore, the “which-way information” can only be meaningfully defined if the context is clarified and the events are explicitly specified in this context.”

Again, I disagree with the choice of the word “events” as I have indicated throughout my review. The authors are referring to “path histories” that are further divided into “sub-histories”. This division, when applied to three-alternative interference experiments like the one described here, admits distinct interpretations about “which-path information” depending on how the division is applied.

- On line 796 in the Methods section, the authors write “After recursive optimisation, the observed visibility between NL1 and NL2 is $98.53 \pm 0.18\%$ and the visibility between NL2 and NL3 is $98.68 \pm 0.17\%$. This indicates that each pair of crystals exhibits a high degree of coherence.” Is there any estimate about what this visibility might be for the third pair, namely, NL1 and NL3? After all, the experiment is performed when all the three crystals are active. If there is any assumption involved here, the authors must explicitly state it.
- I do not find the title of the paper “*Subjective nature of path information in quantum mechanics*” to be adequately justified. In my view, this work points to an inherent ambiguity in the interpretation of path information when applied to three-alternative interference experiments. I don't see this to be a question of subjectivity versus objectivity. If it was subjective, then I would expect that each experimenter would arrive at some definite conclusion about the path information, but different experimenters would be entitled to different conclusions. But that is not the case in this work. Any experimenter would face an ambiguity on the question of path information, and in that sense, it is really an objective ambiguity.
- I don't think that this phenomenon is peculiar to nonlinear interference experiments of the kind described in this work. As such, the ambiguity in path information has nothing to do with two-photon entanglement. Therefore, I expect that this ambiguity would persist even in linear interference

experiments that involve three or more alternatives. For instance, if one could consider one-photon interference in a three-alternative linear Mach-Zehnder interferometer, I would expect that this ambiguity would affect that setup as well. Can the authors comment on this matter?

- In fact, if I may take the previous observation one step further, doesn't the essentially same phenomenon occur even in classical wave interference in a three-alternative Mach-Zehnder interferometer? If one were to set up the interferometer in such a way that the three paths have relative phases $\phi_A = \pi$ and $\phi_C = \pi$, then one could equally consistently interpret the output wave as resulting from a cancellation of wave amplitudes in arm 1 and arm 2, or alternatively in arm 2 and arm 3. I understand that this situation involves a superposition of classical wave amplitudes instead of quantum probability amplitudes, but there is essentially the same ambiguity about whether the energy of the output wave traversed arm-1 or arm-3. Could the authors comment on this as well?

Overall remarks:

I find this work to be quite interesting from a fundamental point of view. It has the potential to spark debates on the nature of path information in quantum interference. I find the manuscript to be very well-written and the experimental observations to be well analyzed and presented. However, I am still unclear about the precise implications of these observations. As I have explained in this review, I don't agree with the interpretations that have been presented by the authors. As a result, I am unable to recommend the publication of this paper in its current form. However, if the authors carefully consider my comments and are able to address my concerns to my satisfaction, I would be willing to reconsider my decision in the next round of review.

Reviewer #2 (Remarks to the Author):

The authors show that in a system of three SPDC processes configured to emit indistinguishable photon pairs, a standard and natural notion of "which path" information is untenable. In particular, at a working point which minimizes interferometric visibility (and therefore maximizes "which path" information) there are two contradictory ways to ascribe the "which path" information to downconversion sources. I could answer "yes" to all the desiderata for publication, so I recommend that the paper eventually be accepted. Just please address the comments below in a minor revision. I waive my right to anonymity in the peer-review process for this manuscript. (Tyler Volkoff, Los Alamos.)

Comments:

1. The abstract reads like a philosophy paper: "this perception of path information is problematic...", "...reshape our understanding of the whole and part in the context of distinguishability." Just use some clear terminology and state your result.
2. "Path distinguishability, in this sense, means the information of which source produced the photon pair."

"Path distinguishability, in the context of frustrated downconversion, means information about which source produced the photon pair."
3. It is crucial to define what is meant by "indistinguishable sources" in the sentence "If the two sources are indistinguishable...". You have already defined path indistinguishability, so define also what you mean by "indistinguishable sources" (e.g., downconversion processes having the same pump, same nonlinearity, etc.). I know that this is discussed in more detail later in the paper, but a note should be made here.

Also, what is the meaning of "perfect interference" in this sentence? I think it means, according to things you already defined, the possibility of $V=1$. If this is what you mean, then please state it.

4. "If there is no interference, it is possible to bet on the outcome of a "which-source" measurement that we know with a higher probability of where the photons come from and vice versa."

"If interference is reduced, complementarity allows for a more profitable bet on the outcome of a "which-source" measurement due to the increased amount of information concerning the source of the photons."

5. "This stems from the different possibilities of partitioning reality into events..."

"This room for interpretation stems from the different possibilities of partitioning reality into events..."

6. "As stated, the amount of "which-source" information and visibility is exclusive due to the complementarity principle."

"As stated, the amount of "which-source" information and visibility exhibit a trade-off relation due to the complementarity principle."

7. Choose one term and be consistent throughout: "which-source" information or "source information"

Choose one term and be consistent throughout: "which-way" or "which path"

8. For the authors consideration: the perturbative description of the state in (1) is correct, and in agreement with the non-perturbative Hamiltonian dynamics of SU(1,1) networks with indistinguishable or partially distinguishable modes (T.J.V. "Relative phase and dynamical phase sensing in a Hamiltonian model of the optical SU(1,1) interferometer"). A non-perturbative description of SU(1,1) networks based on continuous-variable quantum circuits (e.g., Yurke/Klauder in the simplest case of two downconverters) would give results for the visibility that are inconsistent with predictions from the Hamiltonian dynamics at some order in the nonlinearity.

9. "Thus, the rate of emitted photons is given by..."

"Thus, for a fixed relative phase ϕ_A between NL1 and NL2 defining the parameter α , the rate of emitted photons can be rewritten as..."

10. Above (6),

"The rate of emitted photons is given by..."

"For a fixed relative phase ϕ_C between NL2 and NL3 defining the parameter β , the rate of emitted photons can be rewritten as..."

11. In the Discussion, I liked the analogy with a single periodically poled crystal-- regardless of what is happening to the pump phase as it propagates through the crystal, we just keep track of the total downconversion amplitude, not the individual amplitudes from the segments.

12. "...the description of the quantum system must encompass all the possible outcomes, as long as..."

"...the description of a system of quantum emitters (even if it is composed of spatially distinct parts in a laboratory) must encompass all the possible emission outcomes according to the sum of corresponding amplitudes, as long as..."

13. The optical path length difference analyses in the Methods section are important considerations for indistinguishability of the emitted pairs from the three crystals. The authors clearly understand this.

14. The theory of uncertainty relations and wave-particle duality has progressed to much more descriptive and nuanced versions of the $D^2 + V^2 \leq 1$ relation, even for binary interferometers (see, e.g., Coles et al. "Equivalence of wave-particle duality to entropic uncertainty"). Therefore, it is hard to be satisfied by the interpretation of the "which path" information simply in terms of a definite downconversion source-- maybe there is a tradeoff between the visibility in this experiment and a variance or entropy of some measurement that can be considered as a more appropriate notion of "which path" information in the system and which does not lead to any interpretive contradictions. Can the authors provide any insight?

Reviewer #3 (Remarks to the Author):

In this manuscript, the authors consider a frustrated SPDC system with three nonlinear crystals to investigate deducible "source information" when grouping the crystals in different ways. Theoretically, they have observed that: If (1) each crystal emits photon pairs of the same amplitude and (2) the relative phase between two sequential crystals in the photon's path is set at π , the overall photon-pair generation rate of any two sequential crystals combined vanishes. Since we can meet these two conditions simultaneously for both the first & second crystals, as well as the second & third crystals, this appears to create a paradoxical situation where, even though the overall generation rate is nonzero, the photon pairs are seemingly not generated from any pair of the crystals. They use this final observation to conclude that "it is impossible to ascribe a definite physical origin of the photon pair even if the emission probability of one individual source is zero and full path information is available". They have also performed carefully designed experiments to confirm their theoretical findings.

While their observations and findings are thought-provoking, some gaps leading to their conclusion need further clarification. To start with, the whole reasoning seems to rest on the following premise in the abstract: "if visibility is zero, a high path distinguishability can be obtained". However, if we look at the visibility-distinguishability tradeoff given in [4], we see that these quantities satisfy an inequality, rather than an equality. This means that, even in that context, the above premise does not necessarily hold. Although the tradeoff inequality can be saturated for certain pure states, to our knowledge, it has not been established that the inequality holds as an equality for all pure states. Even if the equality does hold for all pure states in an interferometric setup, it is unclear whether this quantitative expression relating visibility and which-way information can be translated into a similar tradeoff between visibility and "which-source" information in the non-interferometric setup considered in this work. Importantly, in the former scenario, the amplitudes and the relative phase are between the two different arms/ kets, which is not the case here. To this end, it is also unfortunate that notions like "path information" and "source information", which are central to the whole argument, were not properly introduced. Consequently, one cannot even readily verify the validity of claims like "full path information is available"; or are we supposed to understand this to mean that the visibility is zero?

Coming back to the observations, if we think of NL1 & NL2 as one source, the authors argued that their combined probability amplitude is zero. While this statement looks innocent, does this mean we are entitled to think of

"combined probability amplitude of A and B is zero = the *emission* probability of both A and B as zero"?

From Equation (1), there seems to be no legitimate reason to consider only the probability from summing the amplitude of two of the three terms, all associated with the *same* ket vector $|s\rangle|i\rangle$. Even if the above statement is valid, when $A = NL1$ & $B = NL2$, one must conclude that NL3 must have contributed to generating the photon pairs in this case. Similarly, if $A = NL2$ & $B = NL3$, one must conclude that the photon pairs are generated by NL1. This seems to be the basis for the authors' claim that we now have two "contradictory which-way information", when both $A = NL1$ & $B = NL2$, and $A = NL2$ & $B = NL3$ conditions are satisfied. However, at least three arguments tell us that such reasoning might not be proper, or must be trodden on carefully.

First, if we push this view further and think that interaction between the fields of the signal/ idlers from NL1 & NL2 happens before the interaction with NL3—after all, the laser field first passes NL1 before NL2, and finally NL3—then one may argue that the NL1 & NL2 combined emission probability amplitude must be zero first, while the non-zero probability emerges later after the laser hits NL. So, whatever is detected must be emitted from NL3. Then, visibility is zero, source knowledge = NL3, and so there's no ambiguity of "which-source"

knowledge; everything appears consistent.

Second, what prevents the possibility that *some* of the photon pairs are from NL1 & NL2, and some from NL2 & NL3? This point of view is consistent with the experimental results as well.

Third, recall Feynman's important lesson about quantum theory: we have to add all the relevant amplitudes before computing the probability. So, it may simply be incorrect to think of zero emission probability of both NL1 & NL2 (or NL2 & NL3), as this possibility happens, via Equation (1), in superposition with the emission from NL3 (or NL1).

All these “inconsistencies” point to another potential pitfall in concluding from quantum mechanical predictions using classical “intuitions”. Presumably, this is the basis for the following passage in the abstract “our findings shed new light on the physical interpretation of probability assignment and path information beyond its mathematical meaning and reshape our understanding of the whole and part in the context of distinguishability.” However, is it not also evident from the first paragraph of the Discussion and conclusion that the lesson to be learned here is, to some extent, already known in [8] (published more than 30 years ago) and [19] (in the PhD thesis of one of authors, published more than 4 years ago). If so, in what way has this shed new light on our understanding?

To summarize, while we agree that this work is thought-provoking and can potentially help us appreciate better the role of interference in a “which-source” setup, its current presentation does not make a strong case for publication in Nature Communications. Besides the issues raised above, the authors can also help improve the clarity of the work by addressing the following issues:

Lines 323/343: The authors should explain clearly what constitutes a 4f system. The descriptions around Line 323 and those around Line 343 are not consistent. Is it defined by lenses between each pair of crystals (as described in the former), or is it a pair of crystals (as described in the latter)?

Lines 399 & 401: “canned” should have been “scanned”

Line 417, 741-742, 745, 762: The hyperlinks for Extended Data Fig. 1b, Extended Data Fig. 1, Extended Data Fig. 2, and Extended Data Fig. 3 do not link the Figures under Extended Data, but to the main text.

Lines 497-498: From the paragraph after Eq. (4), it seems like the off (on) status corresponds to one setting the amplitude to be equal in magnitude and the relative phase to π (or not). However, from the description given here, the phases are both set to π . Then, what distinguishes the on/ off status of the crystals NL1 or NL3?

Lines 485-505: Following the last comment, the coincidence count rates $CC_{\{NL3\}}$ and $CC_{\{NL1\}}$ were presumably carried out under different experimental settings (with different crystals turned off). If so, what entitles us to compare and even combine the probability of photon pairs generated in the individual crystals? And why should one expect that this has anything to do with the situation when all crystals are turned on?

Line 1048: Did the authors mean to write $\phi_A = \pi$ or $\phi_C = \pi$?

Line 1072: How are the counts of Extended Data Fig. 2 normalized for different choices of ϕ_A ? And why are the curves for $\phi_A=0$ and $\phi_A=2\pi$ different?

Line 1214: It would be helpful to indicate either on the images or in the caption which images are for the crystal plane and which others are for the Fourier plane.

Reviewer #4 (Remarks to the Author):

Author Responses to Initial Comments (NCOMMS-25-37771-T):

We are grateful to the Reviewers for supporting this review process and allowing us to explain and, where necessary, refine our arguments through their assessments. We hope we have implemented the suggested points and clearly elaborated our arguments, leading to an overall improvement of the manuscript.

What follows is a point-by-point response to the Reviewers' comments. The following colour code is used:

- Black: Reviewers' comments.
- Blue: Responses to Reviewers' comments.
- Red: Changes in the revised manuscript.

Reviewer #1 (Remarks to the Author):

I have carefully read the manuscript titled "*Subjective nature of path information in quantum mechanics*". In this work, the authors describe a quantum optical interference experiment in which the which-path question appears to admit two distinct but equally self-consistent answers. In what follows, I outline my queries and comments regarding this work:

We thank the Reviewer for their time and valuable comments on our manuscript. We are very pleased that the Reviewer has thoroughly read our manuscript and given these insightful comments. This provides us a good opportunity to illustrate our work more deeply and we are pleased that this has intrigued so many valuable discussions before its final publication. Some of the wording in our results was somewhat misleading, and we are glad to have the opportunity to clarify it in the revised manuscript.

R1.1 There appears to be a contradiction between

sentence (line 38) "*We demonstrate the simultaneous observation of zero interference visibility and the complete absence of which-path information using a three-crystal interference setup.*"

and

sentence (line 123) "*Unlike the two-crystal case, ascribing a definite origin to a photon pair is impossible even though full "path information" is available and no interference is observed.*"

The authors may please correct this discrepancy.

We thank the Reviewer for pointing out this discrepancy. Note that there are different meanings of "path information" in "*the complete absence of which-path information*" in the first sentence and "*full "path information" is available*" in the second sentence. The "path information" in the first sentence means one doesn't know where the photon originates when **one treats the three crystals as a whole system and traces back to the photon's origin**. Then, "path information" in the second sentence means that one knows where the photon originates when two crystals are grouped together and the result is interpreted. Note also that the double quote in the "path information" of the second sentence indicates a different meaning there. We understand that this may cause some confusion because we unintentionally use the same word in different contexts.

To address this confusion, we revise the sentences of the main text:

Line 037: "*We investigate the complementarity between path information and interference visibility through an experiment involving three sources emitting into identical modes. Our findings challenge the classical intuition that a particle can be traced back to its origin through its trajectory when full path information is available.*"

Line 126: “Unlike the two-crystal case, ascribing a definite origin to a photon pair is impossible even though full “path information” is available and no interference is observed if two crystals were grouped together.”

R1.2 On line 72, the word “outcomes” may be replaced by “path histories”. Also, “unpredictable” does not seem fully accurate because this is a statement about retrodiction and not prediction. Perhaps “unascertainable” could be a better choice.

We thank the Reviewer for suggesting this better wording. As also pointed out in the following comments, “path alternatives” or “path histories” are appropriate for our case. We have reworded the sentences to yield:

Line 072: “Conversely, if interference fringes are observed with high visibility, no meaningful path information can be obtained, and path alternatives become entirely unascertainable.”

R1.3 On line 100, the sentence reads “Frustrated down-conversion thus provides a platform to study the interplay between indistinguishability, interference, and which-source information.” Perhaps the word “interference” may be redundant here because the tradeoff is between indistinguishability and which-path information.

A good suggestion. The indistinguishability/distinguishability has a similar meaning to which-source information. If it is distinguishable, then one has the which-source information; vice versa. If it is indistinguishable, then one does not have the which-source information; vice versa.

To address it, we rewrite the sentence as follows:

Line 108: “Frustrated down-conversion thus provides a good platform to study the interplay between interference and which-source information.”

R1.4 On line 118, the sentence reads “This stems from the different possibilities of partitioning reality into events that are represented by probability amplitudes.” As the term “event” refers to something that happens at a given location in space and instant in time, it does not seem appropriate here. I think the word “events” may be replaced by “path histories” for accuracy.

We thank the Reviewer for this suggestion. In the same spirit as R1.2, R1.10, and the following comments, “path alternatives” or “path histories” are a good choice for us. To be consistent, we changed it to “alternatives”.

Line 121: “This room for interpretation stems from the different possibilities of partitioning reality into alternatives that are represented by probability amplitudes.”

R1.5 On line 223, “Now, we can apply the duality relation between the observed visibility” may be rephrased to “Now, one can apply the duality relation between the observed visibility”. Similarly, in line 278 “Thus, we can conclude that the photons were emitted by S1’, that is, NL1.” may be rephrased to “Thus, one can conclude that the photons were emitted by S1’, that is, NL1.”

Thank the Reviewer for this beautiful phrasing. We gratefully accept the proposed change of wording.

Line 243: “Now, one can apply the duality relation between the observed visibility ...”.

Line 302: “Thus, one can conclude that the photons were emitted by S1’, that is, NL1.”.

R1.6 On line 317, the sentence reads “Phase plates are placed between each pair of crystals and are used to tune the relative phase (ϕ_A and ϕ_C) between the pump, signal, and idler photons.” In my view, the phrase “between the pump, signal, and idler photons” is vague. The quantities ϕ_A and ϕ_C are

relative phases between probability amplitudes corresponding to the path alternatives of the process from the source to the detector.

We thank the Reviewer for this remark. More accurately speaking, the three probability amplitudes correspond to three alternative processes from the pump laser to the crystal (NL1, NL2, NL3) and to the detector, as illustrated in Fig.~1 of comment R1.11. In this sense, the quantities ϕ_A and ϕ_C are relative phases between these probability amplitudes. However, this is from the outcome's perspective, because one only gets these three terms and applies a relative phase to them. In the experiment, one needs to adjust the phase. One can see that, during each process, the pump propagates from the pump laser to each crystal and accumulates phase φ_p . After the SPDC process in each crystal, the signal and idler photons begin to propagate from each crystal to the detector and accumulate phase φ_s and φ_i , respectively. The final phase preceding each term is also the relative phase among the phase differences accumulated by the pump beam φ_p , signal φ_s and idler φ_i between each two emission processes. This has also been clearly presented in the literature of nonlinear interferometers, e.g., Eq. (6) in [Chekhova and Ou, Adv. Opt. Photon. 8, 104 (2016)] and Eq. (4) of the seminal work of frustrated down-conversion in [Herzog et al., Phys. Rev. Lett. 72, 629 (1994)]. Therefore, from the experiment perspective, we need to tune the relative phases between the pump, signal, and idler photons.

To refine the meaning, we adapted the sentence to:

Line 340: "Phase plates are placed between each pair of crystals and are used to tune the relative phase between the pump, signal, and idler photons, and thus changing the relative phase ϕ_A and ϕ_C ."

R1.7 In the caption for Fig 3, there are multiple typos. Please correct them.

Thank the Reviewer for pointing out these detailed errors. This is also pointed out by Reviewer #3. We have corrected these multiple typos and highlighted them in the following:

Line 438: "Fig. 3 Three-**crystal** interference obtained with a 2D scan. **a**, Contour plot shows the detected coincidence counts (CC) when all three crystals are active upon varying both phases ϕ_A between NL1 and N2, and ϕ_C between NL2 and NL3. **b**, Visibility observed between S1 and S2 when each phase ϕ_A is fixed and phase ϕ_C is **scanned**. The inset shows the first perspective, which **regards** the system as a two-process interferometer between S1 and S2. At the phase point $\phi_A = (2k + 1)\pi$ ($k \in \mathbb{Z}$), the photon pair emission from S1 is suppressed, as shown schematically in d. **c**, Visibility observed between S1' and S2' when each phase ϕ_C is fixed and phase ϕ_A is **scanned**. The inset shows another perspective, which **regards** the system as a two-process interferometer between S1' and S2'. In an analogous manner, the photon pair emission from S2' is suppressed at the phase point $\phi_C = (2k + 1)\pi$ ($k \in \mathbb{Z}$), as shown schematically in e. The coincidence count data in frames I, II, III of b,c are shown in Fig. 4a,b, respectively. The green line in b and c is the theoretical prediction. The blue dot is the experimental data. The dashed orange line is a guide for the eye. BS: Beam Splitter."

R1.8 On line 455, the sentence reads "In our experiment, the visibilities of the interference between S1 and S2 and between S1' and S2' when setting $\phi_A = \pi$ and $\phi_C = \pi$ are $9.12 \pm 3.81\%$ and $8.30 \pm 4.19\%$, respectively." Here S1, S2, S1', S2' refer to effective sources. Strictly speaking, the interference is not between sources but rather between probability amplitudes corresponding to downconversion happening at those sources. Although I agree that brevity and simplicity may warrant some flexibility, the authors may want to explore other rephrasings. Also, do these visibilities follow any duality relation or tradeoff with respect to the path probabilities p_3 and p_1 calculated on line 492-294? If yes, I would suggest that the authors point this out.

We thank the Reviewer for this comment. This is basically the same problem as comment R1.11. We agree that the interference is between probability amplitudes corresponding to different alternatives happening at each source. As argued in the answer to comment R1.11, each source contributes to a probability amplitude. That's because our interferometer is a nonlinear interferometer. Essentially, for this type of interferometer, interference happens between the different alternatives a photon pair can be generated, each leading to a detection in the same mode.

Yes, the Reviewer is insightful. These visibilities follow the duality relation with respect to the path probabilities p_3 and p_1 calculated on line 524. Thank you for providing this opportunity to present this information to the reader. In the following, we gave the duality relation. One can see $D^2 + V^2 < 1$, but it is also close to 1, which means one can use the duality relation to determine with high probability where the photons originate when perceiving two sources as one.

To clarify the interference, we revised the sentence:

Line 505: “In our experiment, the visibility of the interference between the probability amplitude contributed by S1 and that of S2 when setting $\phi_A = \pi$ is $9.12 \pm 3.81\%$. The corresponding visibility between S1' and S2' when setting $\phi_C = \pi$ is $8.30 \pm 4.19\%$.”

To give the duality relation, we listed a table in the Supplementary Information:

Supplementary Table A1: Experimentally obtained duality relations.

	V	$D = p_3^2 / p_1^2$	V^2	D^2	$D^2 + V^2$
S1&S2	0.0912	0.9514	0.008317	0.905162	0.913479
S1'&S2'	0.0830	0.9641	0.006889	0.929489	0.936378

R1.9 On line 493, the sentence in the parentheses “(the probability is calculated by CC_{NL1}/CC_{tot} and CC_{NL3}/CC_{tot} , where $CC_{NL1} = CC_{tot} - CC_{S2'}$ and $CC_{NL3} = CC_{tot} - CC_{S1}$, $CC_{S1}/CC_{S2'}$ are counts when crystal NL3/NL1 is off and the phases are set to π , CC_{tot} are the counts when all three crystals are on and the phases are set to π)” is confusing. It is not clear which quantity refers to p_1 and which refers to p_3 . Please specify explicitly. Also, when the authors say “the phases are set to π ”, they are probably referring to ϕ_A and ϕ_C , but it is better to specify as “both phases ϕ_A and ϕ_C are set to π ”.

We thank the Reviewer for pointing out this confusion. Yes, the phases are referring to both ϕ_A and ϕ_C . To make it clear, we revised the sentence as follows:

Line 524: “(probability p_1 is calculated by CC_{NL1}/CC_{tot} and probability p_3 is calculated by CC_{NL3}/CC_{tot} , where $CC_{NL1} = CC_{tot} - CC_{S2'}$ and $CC_{NL3} = CC_{tot} - CC_{S1}$, CC_{S1} and $CC_{S2'}$ are counts when the crystal NL3 and NL1 are off, respectively, and the corresponding phase ϕ_A and ϕ_C is set to π , CC_{tot} are the counts when all three crystals are on and both phases ϕ_A and ϕ_C are set to π)”

R1.10 On line 524, the authors write “However, the assignment of a probability amplitude of zero to an event should not be interpreted as a zero probability that this event will occur.” As I have pointed out before, the use of the word “event” is problematic. The authors are referring to an entire “path history” or “path alternative” from the pump source to the detector. The probability amplitudes correspond to these alternatives, and not to specific events.

We thank the Reviewer for this helpful remark. Again, the same problem also exists in comments R1.4 and R1.15; this comment is also closely related to comment R1.11. In answer to comment R1.11, we show the three “process alternatives” from the pump laser to the detector. We agree with the Reviewer that the probability amplitude corresponds to these alternatives. We have changed the phrasing which reads:

Line 556: “However, the assignment of a probability amplitude of zero to an alternative should not be interpreted as a zero probability that this alternative will occur.”

R1.11 On line 526, the authors write “In the experiment of frustrated down-conversion [8], ideally no

photon pairs are detected if there is complete destructive interference between two photon-pair emission processes. Nevertheless, the possibility for a photon pair originating from frustrated down-conversion cannot be ruled out if a third nonlinear crystal were inserted before the detectors. Therefore, a measured probability of zero for an isolated subsystem does not mean that this probability remains zero in the presence of other subsystems [19], since all subsystems can interfere with each other.”

I disagree with the above reasoning. When an additional crystal is added to the system, the system has changed in a non-trivial manner. There is no reason to expect that the photon pair rate would remain zero. Mathematically speaking, the total probability amplitude for the entire process must now have an additional term. Contrary to what the authors say, the sum of the two original terms still remains zero. It is the introduction of the new term corresponding to the third crystal that will now make the total photon pair rate non-zero. There is nothing confusing or surprising about this. The situations with and without the third crystal are two completely different situations.

We agree with the Reviewer’s comment. Actually, what we want to express is that the situation will be different when a third crystal is inserted. Therefore, we cannot talk about the probability of the original two terms anymore, because all three crystals are now in action. We must consider all the contributions from the three alternatives. To clarify, we have rephrased the sentences of our reasoning.

Basically, we can’t see any disagreement between the Reviewer’s comment and our reasoning. Physically speaking, the system has changed in a non-trivial manner, so as the Reviewer said, “*There is no reason to expect that the photon pair rate would remain zero.*” However, the Reviewer also said, “*Contrary to what the authors say, the sum of the two original terms still remains zero.*” This is from a mathematical perspective. These reflect how one interprets the probability assignment beyond its mathematical meaning, as stated in our abstract. Mathematically, if $|A + B|^2 = 0$, we can still say $A + B = 0$ in $|A + B + C|^2$. However, the isolated terms of the quantum state do not allow a physical interpretation. Physically, we cannot say anything about the probability of the two original terms, as all three indistinguishable terms must be considered in the total probability. The introduction of the new term corresponding to the third crystal makes the total photon pair rate non-zero. Nevertheless, it also produces something very different from the two-crystal case. In the two-crystal case, if one crystal is switched off and another is added, non-zero photon counts are observed, and it can be determined with high probability that the photons originate from the second crystal. Here, the physical interpretation is consistent with the mathematical formulation. For the three-crystal case, the addition of the third crystal makes the counts non-zero, but one can’t determine where the photons originate from.

Also, it is not accurate to say “subsystems can interfere with each other” because interference does not happen between systems (or subsystems) per se. This issue was at the heart of Dirac’s remark that a photon only interferes with itself, and not with any other photon. Roy Glauber, for instance, later clarified that interference happens between probability amplitudes corresponding to path histories or path alternatives. And these alternatives are specified all the way from the original source to the detection stage. This is the basic premise of even the path integral formalism pioneered by Feynman.

Thank the Reviewer for bringing this issue to our attention. We understand the Reviewer’s concern. In fact, this comment is related to comments R1.12 and R1.13. In our view, interference happens between probability amplitudes corresponding to different alternatives. These alternatives can refer to paths of a particle or to any other potential process that leads to a specific outcome. In our case, the alternatives refer to a pump photon being down-converted in the i -th crystal and the resulting photon pairs arriving at the detectors. To also respond to R1.13, one can treat the system as a whole by considering alternative processes that occur within each crystal due to second-harmonic generation. These alternative processes are shown in Fig.~1. The first process I is from the pump laser to the crystal NL1 to the detector. The second process II is from the pump laser to the crystal NL2 to the detector. The third process III is from the pump laser to the crystal NL3 to the detector. Each crystal contributes to each alternative, e.g., NL1 contributes to the probability amplitude $a e^{i\phi_A} |s\rangle |i\rangle$, NL2 contributes to the probability amplitude $b |s\rangle |i\rangle$, and NL3 contributes to the probability amplitude $c e^{i\phi_C} |s\rangle |i\rangle$.

Fig. 1 Interference of three processes.

To account for this comment, we rewrite the sentence:

Line 558: “In the experiment of two-crystal frustrated down-conversion [8], if one crystal emits no photons, one can determine with high probability that the detected photons originate from the second crystal. Nevertheless, the probability must include all the contributions of the three crystals if a third nonlinear crystal were inserted before the detectors. Different from the two-crystal case, a measured probability of zero for the first two crystals does not mean that no photons come from them in the presence of a third crystal, since each crystal contributes to a probability amplitude alternative, which finally interferes with each other [19].”

R1.12 On line 561, the authors write “The one-particle and two-particle interference are widely characterized in the framework of the duality relation. This interference phenomenon has its roots in the fundamental indistinguishability of identical quantum particles.”

I disagree with this remark. The origin of interference is not the indistinguishability or the identicalness of the particles themselves, but rather of the alternatives or the path histories of the system as a whole. That system could be a one-particle system or a two-particle system depending on the kind of experiment being performed.

Note that if what the authors state was true, it should never be possible to observe two-photon interference effects with non-degenerate entangled photons from parametric down-conversion. However, Hong-Ou-Mandel-type effects and other two-photon interference effects have been observed with two-photon systems comprising of non-identical photons. In fact, the famous postponed compensation experiment reported in the paper titled “Can two-photon interference be considered the interference of two photons?” *Phys. Rev. Lett.* **77**, 1917 (1996) shows a Hong-Ou-Mandel-type effect with the photons not even appearing at the beam-splitter at the same time. Such experiments clearly demonstrate that it is not the indistinguishability or identicalness of the photons themselves that counts, but rather of the alternatives of the evolution of the system as a whole.

Many thanks to the Reviewer for pointing out this problem. We completely agree that the indistinguishability of ‘alternatives’ or ‘path histories’ can be seen as the origin of interference phenomena, and thank the Reviewer for this remark, and have modified the sentences to a more precise wording.

To clarify it, we adapted the sentences as follows:

Line 595: “The one-particle and two-particle interference are widely characterised in the framework of

the duality relation. This interference phenomenon has its root in the indistinguishability of the different alternatives which contribute to the overall quantum amplitudes of the system as a whole.”

R1.13 On line 573, the authors write “Our experiment also demonstrates one important feature of quantum mechanics: the description of the quantum system must encompass all the possible outcomes, as long as no intervention is made to make the parts distinguishable in principle. Whether the quantum system should be treated as a whole or as separable parts depends on whether the parts can be distinguished from each other.”

I don't quite agree with this claim. Strictly speaking, the experiment has only two possible outcomes, namely, either the photon pair is detected or it is not. But the description must encompass all the possible alternatives or path histories, not outcomes. In addition, it is not really a question of making the parts distinguishable, but rather about making the alternatives distinguishable. The quantum system under investigation is the two-photon field, not the crystals. And it is imperative to treat this two-photon system as a whole right from the pump source to the detectors. The present study does not raise any questions about this matter.

We thank the Reviewer for bringing these issues to our attention. The Reviewer is right, the quantum system that is actually detected is the photon pairs, not the crystals. The description must encompass all possible alternatives. However, if it's a question of making the parts distinguishable or making the alternatives distinguishable. In our view, both matters. These issues are also related to comments R1.11, R1.12, and R1.18. We agree that “The description must encompass all possible alternatives”. In fact, whether the alternatives are distinguishable or not is determined both by the experiment arrangement and the properties (intrinsic + extrinsic) of the particles (we have put some discussions in the “Supplementary Information to comment R1.12”). The identity of the particles, i.e., the indistinguishability of the intrinsic properties such as mass, charge, spin, etc, is the premise for the alternative indistinguishability. When some of the extrinsic properties are distinct, the experiment for multi-photon interference is designed in a sophisticated way (detector blind to these properties, post-selection, compensation) to erase the information carried by the individual particles, making the alternatives indistinguishable. However, if any of the distinct information from individual particles is utilised or labelled by the experiment to discern the different alternatives, then no interference will happen. Taking the example of the interference of a photon and an electron in a HOM setup. With a photon and an electron, their fundamental distinguishability immediately labels each particle's path. Even if spatially overlapped at a beam splitter, the electron and photon carry entirely different physical properties, such as mass, charge, spin-statistics, and interactions, that clearly distinguish their alternative quantum amplitudes. Therefore, the indistinguishability of the parts also matters.

In the answer to comment R1.11, we have shown three alternative processes which interfere with each other. We hope this addresses the Reviewer's concern about treating the two-photon system as a whole.

To address these issues and also conform to comment R2.12, we rewrite the text as follows.

Line 609: “Our experiment also demonstrates an important feature of quantum mechanics: the description of the interference of a system of quantum emitters (even if it is composed of spatially distinct parts in a laboratory) must encompass all possible alternatives according to the sum of the corresponding amplitudes, as long as no intervention is made to make these alternatives distinguishable. Whether the three-crystal system should be treated as a whole or can be analysed separately depends on whether their contributions to these different alternatives can be distinguished or not.”

R1.14 On line 586, the authors write “By considering two of the three sources as a single source, two contradictory interpretations of the path information are possible within the experimental configuration.”

I am not sure “contradictory” is the right word here. Within the quantum formalism, there is really no contradiction. There are two distinct common-sense interpretations that are equally self-consistent but mutually incompatible from a common-sense point of view.

Thank you. The interpretations are incompatible, but they lead to a contraction of the mathematical descriptions of the experiment results. To address it, we have reworded it in the places where “contradictory” may not be appropriate:

Line 645: “By considering two of the three sources as a single source, two incompatible interpretations of the path information are possible within one experimental configuration.”

Line 133: “There are multiple valid ways to assign probability amplitudes to the alternatives in the experiment, leading to **incompatible** interpretations of the results.”

Line 588: “all these pictures are equally valid and self-consistent, although they lead to **incompatible** which-path information.”

R1.15 On line 591, the authors write “Usually, we identify macroscopically distinguishable “events” and assign probability amplitudes to them. However, this identification of “events” is generally ambiguous. Any event could in principle be subdivided into subevents of non-zero probability amplitude, which can lead to a zero probability amplitude of the combined event. Therefore, the “which-way information” can only be meaningfully defined if the context is clarified and the events are explicitly specified in this context.”

Again, I disagree with the choice of the word “events” as I have indicated throughout my review. The authors are referring to “path histories” that are further divided into “sub-histories”. This division, when applied to three-alternative interference experiments like the one described here, admits distinct interpretations about “which-path information” depending on how the division is applied.

We thank the Reviewer for this suggestion. Based on the previous arguments, “process alternatives” are better for our setup, while “path alternatives” are appropriate for a traditional linear interferometer. To be consistent and general, we changed it to be “alternatives” and “sub-alternatives”:

Line 648: “Usually, we identify macroscopically distinguishable “alternatives” and assign probability amplitudes to them. However, this identification of “alternatives” is generally ambiguous. Any probability could, in principle, be subdivided into alternatives of non-zero probability amplitude, which can lead to a zero-probability amplitude of the combined probability. Therefore, “which-path information” can only be meaningfully defined if the context is clarified and the alternatives are explicitly specified in this context.”

R1.16 On line 796 in the Methods section, the authors write “After recursive optimisation, the observed visibility between NL1 and NL2 is $98.53 \pm 0.18\%$ and the visibility between NL2 and NL3 is $98.68 \pm 0.17\%$. This indicates that each pair of crystals exhibits a high degree of coherence.” Is there any estimate about what this visibility might be for the third pair, namely, NL1 and NL3? After all, the experiment is performed when all the three crystals are active. If there is any assumption involved here, the authors must explicitly state it.

Our argument is independent of the coherence between photon pair emission from NL1 and NL3. The interference between NL1 and NL2 is used to “switch off” S1 and similarly NL2 and NL3 for S2'. The third possible partition S1" being comprised of NL1&NL3 and S2" being NL2 is never used in our argument.

Nevertheless, we estimated the visibility between NL1 and NL3 for a complete understanding of our experimental system. The estimated visibility for NL1 and NL3 is $V_{13} = 97.24 \pm 0.25\%$. For each pair of sources, the degree of coherence can be used to account for the non-ideal interference.

$$R_{12}(\phi) = A^2 + B^2 + 2AB \cdot |\gamma_{12}| \cdot \cos(\phi)$$

$$R_{23}(\phi) = B^2 + C^2 + 2BC \cdot |\gamma_{23}| \cdot \cos(\phi)$$

A , B and C are amplitudes, which can be obtained from independently measured photon pair rates produced by individual sources. The degree of coherence can be defined as $|\gamma_{12}| = \cos e1$, $|\gamma_{23}| = \cos e3$. Using the counts from each source and visibility $V_{12} = 98.53 \pm 0.18\%$ and $V_{23} = 98.68 \pm 0.17\%$ in the experiment, one can calculate $e1$ and $e3$. Then the quantum state can be constructed as,

$$|\psi\rangle = s1 + s2 \cdot e^{i\phi_1} + s3 \cdot e^{i(\phi_1+\phi_2)}$$

where $s1 = A(\cos e1 |1,0,0\rangle + \sin e1 |0,1,0\rangle)$, $s2 = B|1,0,0\rangle$, $s3 = C(\cos e3 |1,0,0\rangle + \sin e3 |0,0,1\rangle)$ are the states defined in the three-mode Fock space $|1,0,0\rangle, |0,1,0\rangle, |0,0,1\rangle$, which accounts for the non-ideal interference due to some distinguishability in different modes. Using the number operator $\hat{n} = \hat{a}^\dagger \hat{a}$ in the second quantization, one can calculate the count rate $R_{13}(\phi) = \langle \psi | \hat{n} | \psi \rangle$. By assuming $s2 = 0$ and $\phi_1 = 0$, one can estimate the visibility of NL1 and NL3.

To account for this, we complemented some texts in the revised manuscript:

Line 877: "For the source pair NL1&NL2 and NL2&NL3, the count rates can be expressed as in the non-ideal interference,

$$R_{12}(\phi) = A^2 + B^2 + 2AB \cdot \cos e1 \cdot \cos \phi$$

$$R_{23}(\phi) = B^2 + C^2 + 2BC \cdot \cos e3 \cdot \cos \phi$$

A , B and C are amplitudes corresponding to each crystal, $e1$ and $e3$ accounts for the degree of coherence in the non-ideal case. Defining $s1 = A(\cos e1 |1,0,0\rangle + \sin e1 |0,1,0\rangle)$, $s2 = B|1,0,0\rangle$, $s3 = C(\cos e3 |1,0,0\rangle + \sin e3 |0,0,1\rangle)$ in the three-mode Fock space $|1,0,0\rangle, |0,1,0\rangle, |0,0,1\rangle$, the quantum state can be constructed as,

$$|\psi\rangle = s1 + s2 \cdot e^{i\phi_1} + s3 \cdot e^{i(\phi_1+\phi_2)}$$

Then, using the number operator $\hat{n} = \hat{a}^\dagger \hat{a}$, one can estimate the visibility between NL1 and NL3 without NL2, i.e., $s2 = 0$, and assuming $\phi_1 = 0$ without loss of generality. Based on the measured counts and visibility V_{12} and V_{23} , the estimated visibility of NL1 and NL3 is $V_{13}=97.24\pm 0.25\%$. This indicates that each pair of crystals exhibits a high degree of coherence."

R1.17 I do not find the title of the paper "*Subjective nature of path information in quantum mechanics*" to be adequately justified. In my view, this work points to an inherent ambiguity in the interpretation of path information when applied to three-alternative interference experiments. I don't see this to be a question of subjectivity versus objectivity. If it was subjective, then I would expect that each experimenter would arrive at some definite conclusion about the path information, but different experimenters would be entitled to different conclusions. But that is not the case in this work. Any experimenter would face an ambiguity on the question of path information, and in that sense, it is really an objective ambiguity.

We understand the point raised by the Reviewer; however, we do not fully agree. In our view, it's not an ambiguity to interpret the path information. What we mean by the "subjective" here is that the experimenter can conceptualize the experiment differently and they actually arrive at different conclusions when they look at it from a different perspective.

Imagine two Gedankenexperiments, there are two black boxes (BB1, BB2) and a phase shifter ϕ in between these two boxes. For experimenter 1 (E1), NL1&NL2 is placed into BB1 and NL3 is placed into BB2. No photons are emitted from NL1&NL2 because of destructive interference. E1 uses a detector to detect the photons and find that the photon counts remain constant when she/he is changing the phase ϕ . E1 first removes BB1 and finds the counts are the same. She/He then places BB1 in and removes BB2, and finds the counts are zero. What conclusions does E1 get? E1 must get the conclusion that the photons originate from BB2. For experimenter 2 (E2), NL1 is placed into BB1 and NL2&NL3 is placed

into BB2. No photons are emitted from NL2&NL3 because of destructive interference. In the same way, E2 must draw the conclusion that the photons originate from BB1. We can see that, fundamentally, this is the same experiment setup. However, why did the two experimenters arrive at different conclusions? That's because they conceptualize the experiment differently. Note that both of these Gedankenexperiments essentially correspond to the two different perspectives of path information interpretation. That they arrive at different conclusions indicates that the interpretation of path information is subjective, depending on how one conceptualizes the experiment's configuration.

Fig. 2 Two Gedankenexperiments.

Furthermore, imagine that a third experimenter discovers that one black box indeed has two crystals; they then open the black boxes. How will she/he conceptualize the experiment? Well, suppose she/he is a quantum physicist and believes in quantum rules. In that case, she/he will tell us the photons have no definite origin because all three contribute to the probability amplitude alternatives, which are indistinguishable. Or, with an ingenious way, she/he can say the photons come from the system as a whole. We can see that there is no ambiguity here. The argument is analogous to Schrödinger's cat. It's not ambiguous whether the cat is alive or dead. It's just that the cat is in a superposition state. Only when the box is opened can we say it's in a definite state. Similarly, one can only definitively say that the photons originate from the third crystal if one truly detects the photons from the first two crystals (e.g., by placing a beam splitter after the second crystal and detecting at one of the output ports). But this has completely changed the setup.

R1.18 I don't think that this phenomenon is peculiar to nonlinear interference experiments of the kind described in this work. As such, the ambiguity in path information has nothing to do with two-photon entanglement. Therefore, I expect that this ambiguity would persist even in linear interference experiments that involve three or more alternatives. For instance, if one could consider one-photon interference in a three-alternative linear Mach-Zehnder interferometer, I would expect that this ambiguity would affect that setup as well. Can the authors comment on this matter?

We thank the Reviewer for putting forward these insightful thoughts, including the next comment. Note that the meaning of path information in our three-crystal interferometer and the three-path linear Mach-Zehnder interferometer (**MZI**) is very different. In the MZI, the path information refers to the path the photon passes through. In the three-crystal interferometer, the path information refers to the place where the photon is generated. In the one-photon three-path linear MZI, the **one photon** must pass through all three paths, even though one blocks C and then blocks A&B when trying to regard A&B as

path S1 and C as another path S2 (see Fig.~3). Because there is only one photon, one-third of the probability that the photon passes through each path, e.g., the state is written as $|\psi\rangle = \frac{1}{\sqrt{3}}(e^{i\varphi_A}|0\rangle + |1\rangle + e^{i\varphi_C}|2\rangle)$. One cannot say this photon only passes through C and not through A and B; otherwise, how can one prove that A and B have a destructive interference if it only passes through C? However, with the three-crystal case, one can conclude that the detected photon pairs originate from crystal C=NL3 because it is reasonable that the detected photons are only generated by C and travel to the detectors when A and B are blocked.

Fig. 3 Three-path MZI.

In fact, this and the next comments also pose the fundamental question: What are the differences between the nonlinear interferometer (an interferometer composed of parametric amplifiers) invested here and the traditional linear interferometer (an interferometer composed of beam splitters), such as the Mach-Zehnder interferometer (**MZI**)? As already proposed and thoroughly analysed by [Yurke et al., Phys. Rev. A 33, 4033 (1986)], the **nonlinear interferometer** incorporates a SU(1,1) transformation that describes the evolution under parametric amplification, while the traditional **MZI** is governed by the SU(2) transformation, which is an evolution of two-mode linear optics. In addition, SU(1,1) transformations describe **active** optical dynamics, which produce a non-classical Gaussian state of light inside the SU(1,1) interferometer, even from **vacuum** input. From the perspective of state evolution, the photon number is conserved in a traditional **MZI**; however, the photon number is not conserved due to amplification in a **nonlinear interferometer**, and it also involves squeezing and entanglement generation. Other literature that compares the differences can also be found in [Hudelist et al., Nature Commun. 5, 3049 (2014); arXiv2505.15635 (2025)].

Thus, we can interpret the traditional **MZI** in two dual ways: (1) Path interference (path histories or path alternatives, as often mentioned by the Reviewer): A photon can take two paths, and the outputs result from the quantum superposition of these paths, which is a particle picture. (2) Field interference (as mentioned by the Reviewer's comment R1.19): Optical fields split and recombine, and their amplitudes interfere, which is a wave picture. However, in the **nonlinear interferometer**, the interference occurs between "the vacuum evolving through the second crystal" and "the amplified field coming from the first crystal" [Yurke et al., Phys. Rev. A 33, 4033 (1986)]. In this sense, it's not just an interference of different path alternatives, but an interference between two processes, i.e., coherent superpositions of different pair-generation amplitudes. Therefore, we can refer to it as the interference between probability amplitudes corresponding to process alternatives. This process interference was even demonstrated in the experiment, when the two processes are distantly separated [Pseiner et al., Phys. Rev. Res. 6, 013294 (2024)]. Note that this is not two-photon interference, despite having two photons.

With the above knowledge, two aspects can be put forward. First, **there is no photon origin problem for one-photon interference in a three-alternative MZI** because there is no photon generation process. Note that the path information we are discussing in this work refers to the trajectory along

which we can trace back to where the photon was born. With one-photon interference in a three-alternative MZI, a photon originates only from the input port of the MZI; therefore, we know its source. Second, **no grouping can be interpreted in the one-photon three-alternative MZI**. Imagine three paths A, B, and C, one group the A and B together and tuning the relative phase between A and B to be π , can we say that the photon only passes through C, not through A and B? Definitely not. It's only when the photon also takes the paths through A and B that we can say there is destructive interference; otherwise, how can the destructive interference from A and B come from? However, in a three nonlinear crystal interferometer, A and B crystals can be grouped together to interpret, as they have their own photon generation process. When these two processes from A and B have destructive interference, we can say that the photon originates from crystal C. This is definitely different from the one-photon three-alternative MZI.

R1.19 In fact, if I may take the previous observation one step further, doesn't the essentially same phenomenon occur even in classical wave interference in a three-alternative Mach-Zehnder interferometer? If one were to set up the interferometer in such a way that the three paths have relative phases $\phi_A = \pi$ and $\phi_C = \pi$, then one could equally consistently interpret the output wave as resulting from a cancellation of wave amplitudes in arm 1 and arm 2, or alternatively in arm 2 and arm 3. I understand that this situation involves a superposition of classical wave amplitudes instead of quantum probability amplitudes, but there is essentially the same ambiguity about whether the energy of the output wave traversed arm-1 or arm-3. Could the authors comment on this as well?

Fundamentally, this comment has been addressed in the previous comment, R1.18. In the classical description, there is no concept of path information, as only waves are used to describe the system. Therefore, one cannot construct the same argument about incompatible interpretations of path information. Note that for the case of frustrated SPDC, a discussion about whether or not the effect could be explained classically ended in a demonstration that the achievable visibility in the classical case is much lower than observed in the experiment [Senitzky, I. R. "Classical interpretation of "Frustrated two-photon creation via interference" Physical Review Letters 73.22 (1994): 3040. Herzog, T., et al. "Herzog et al. reply." Physical Review Letters 73.22 (1994): 3041.]

For the classical waves, one can ask in which path the energy of the classical wave traverses. It will always be 1/3 in the "single" path and 2/3 in the "combined" path. Any cancellation happens when the waves are recombined and in the order in which they are recombined.

In short, classical wave interference is an interference of classical wave amplitudes instead of the quantum probability amplitudes, as the Reviewer pointed out. Therefore, we can only interpret it with the wave picture, not even with a particle picture, as in the one-photon MZI. From this perspective, classical waves can be treated as an electromagnetic field which splits through all three paths and recombines and interferes. No grouping can be made here to interpret, as the field is an entity. It's only **mathematically** that one can say **wave amplitude components** in arm 1 and 2, or arm 2 and 3, cancel. However, the mathematical meaning does not necessarily indicate its physical meaning.

Overall remarks:

R1.20 I find this work to be quite interesting from a fundamental point of view. It has the potential to spark debates on the nature of path information in quantum interference. I find the manuscript to be very well-written and the experimental observations to be well analyzed and presented. However, I am still unclear about the precise implications of these observations. As I have explained in this review, I don't agree with the interpretations that have been presented by the authors. As a result, I am unable to recommend the publication of this paper in its current form. However, if the authors carefully consider my comments and are able to address my concerns to my satisfaction, I would be willing to reconsider my decision in the next round of review.

We thank the Reviewer for their valuable comments, observations, and insights on our manuscript. We hope that our revisions, especially the clarification of fundamental reasons for the interference, the difference with the linear MZI interferometer and the explanations of the subjectivity, have adequately

addressed the Reviewer's concerns.

Reviewer #2 (Remarks to the Author):

The authors show that in a system of three SPDC processes configured to emit indistinguishable photon pairs, a standard and natural notion of "which path" information is untenable. In particular, at a working point which minimizes interferometric visibility (and therefore maximizes "which path" information) there are two contradictory ways to ascribe the "which path" information to down conversion sources. I could answer "yes" to all the desiderata for publication, so I recommend that that the paper eventually be accepted. Just please address the comments below in a minor revision. I waive my right to anonymity in the peer-review process for this manuscript. (Tyler Volkoff, Los Alamos.)

We are very pleased with this general assessment of our work and thank the Reviewer for these kind opening remarks. We are grateful that this Reviewer understands the fundamental difference between a nonlinear interferometer, as interference of three SPDC processes, and a traditional linear interferometer. Thank you for providing improved phrasings of some sentences and constructive comments. The Reviewer makes a good point in relating the duality relation to the entropic uncertainty relation.

Comments:

R2.1 The abstract reads like a philosophy paper: "this perception of path information is problematic...", "...reshape our understanding of the whole and part in the context of distinguishability." Just use some clear terminology and state your result.

Thank you. Indeed, we believe our work may provoke discussions on some philosophical topics. This is beneficial for the broad audience. Therefore, we use a philosophical tone to describe some implications. To alleviate the tone, we revised it as:

Line 037: "We investigate the complementarity between path information and interference visibility through an experiment involving three sources emitting into identical modes. Our findings challenge the classical intuition that a particle can be traced back to its origin through its trajectory when full path information is available. By grouping the crystals in different ways, we demonstrate that it is impossible to ascribe a definite physical origin to the photon pair, even if the emission probability of one individual source is zero and full path information is available. Our results shed new light on the physical interpretation of probability assignment and path information beyond its mathematical meaning and show that the interpretation of path information in quantum mechanics is subjective."

R2.2 "Path distinguishability, in this sense, means the information of which source produced the photon pair."

"Path distinguishability, in the context of frustrated down conversion, means information about which source produced the photon pair."

Revised. Line 101: "Path distinguishability, in the context of frustrated down conversion, means information about which source produced the photon pair."

R2.3 It is crucial to define what is meant by "indistinguishable sources" in the sentence "If the two sources are indistinguishable...". You have already defined path indistinguishability, so define also what you mean by "indistinguishable sources" (e.g., downconversion processes having the same pump, same nonlinearity, etc.). I know that this is discussed in more detail later in the paper, but a note should be made here.

Also, what is the meaning of "perfect interference" in this sentence? I think it means, according to

things you already defined, the possibility of $V=1$. If this is what you mean, then please state it.

Revised. Line 091: "If the two sources are indistinguishable, which means SPDC photons from each source have the same spatiotemporal modes, same frequency and bandwidth, etc., perfect interference can occur, i.e., the interference visibility can reach 1."

R2.4 "If there is no interference, it is possible to bet on the outcome of a "which-source" measurement that we know with a higher probability of where the photons come from and vice versa."

"If interference is reduced, complementarity allows for a more profitable bet on the outcome of a "which-source" measurement due to the increased amount of information concerning the source of the photons."

Revised. Line 103: "If interference is reduced, complementarity allows for a more profitable bet on the outcome of a "which-source" measurement due to the increased amount of information concerning the source of the photons."

R2.5 "This stems from the different possibilities of partitioning reality into events..."

"This room for interpretation stems from the different possibilities of partitioning reality into events..."

Revised. Line 121: "This room for interpretation stems from the different possibilities of partitioning reality into alternatives that are represented by probability amplitudes."

R2.6 "As stated, the amount of "which-source" information and visibility is exclusive due to the complementarity principle."

"As stated, the amount of "which-source" information and visibility exhibit a trade-off relation due to the complementarity principle."

Revised. Line 175: "As stated, the amount of "which-source" information and visibility exhibit a trade-off relation due to the complementarity principle."

R2.7 Choose one term and be consistent throughout: "which-source" information or "source information"

Choose one term and be consistent throughout: "which-way" or "which path"

Thank you. We choose "which-source" information and "which-path", and keep consistent throughout the revised manuscript.

R2.8 For the authors consideration: the perturbative description of the state in (1) is correct, and in agreement with the non-perturbative Hamiltonian dynamics of $SU(1,1)$ networks with indistinguishable or partially distinguishable modes (T.J.V. "Relative phase and dynamical phase sensing in a Hamiltonian model of the optical $SU(1,1)$ interferometer"). A non-perturbative description of $SU(1,1)$ networks based on continuous-variable quantum circuits (e.g., Yurke/Klauder in the simplest case of two downconverters) would give results for the visibility that are inconsistent with predictions from the Hamiltonian dynamics at some order in the nonlinearity.

We thank the Reviewer for this valuable information. We are glad that the description of the state (1) is in agreement with the Hamiltonian-based models of the $SU(1,1)$ interferometer. This again strengthens the argument for the fundamental difference between the nonlinear interferometer and the traditional MZI interferometer. To emphasise this, we cited the relevant references.

Line 164: "This state is in agreement with the Hamiltonian-based models of the SU(1,1) interferometer [20], which is fundamentally different from the traditional Mach-Zehnder interferometer."

R2.9 "Thus, the rate of emitted photons is given by..."

"Thus, for a fixed relative phase ϕ_A between NL1 and NL2 defining the parameter α , the rate of emitted photons can be rewritten as..."

Revised. Line 233: "Thus, for a fixed relative phase ϕ_A between NL1 and NL2, defining the parameter α , the rate of emitted photons can be rewritten as ..."

R2.10 Above (6),

"The rate of emitted photons is given by..."

"For a fixed relative phase ϕ_C between NL2 and NL3 defining the parameter β , the rate of emitted photons can be rewritten as..."

Revised. Line 285: "For a fixed relative phase ϕ_C between NL2 and NL3, defining the parameter β , the rate of emitted photons can be rewritten as ..."

R2.11 In the Discussion, I liked the analogy with a single periodically poled crystal-- regardless of what is happening to the pump phase as it propagates through the crystal, we just keep track of the total downconversion amplitude, not the individual amplitudes from the segments.

We thank the Reviewer for this positive remark.

R2.12 "...the description of the quantum system must encompass all the possible outcomes, as long as..."

"...the description of a system of quantum emitters (even if it is composed of spatially distinct parts in a laboratory) must encompass all the possible emission outcomes according to the sum of corresponding amplitudes, as long as..."

Revised, also based on the comment R1.13. Line 609: "Our experiment also demonstrates an important feature of quantum mechanics: the description of the interference of a system of quantum emitters (even if it is composed of spatially distinct parts in a laboratory) must encompass all possible alternatives according to the sum of the corresponding amplitudes, as long as no intervention is made to make these alternatives distinguishable. Whether the three-crystal system should be treated as a whole or can be analysed separately depends on whether their contributions to these different alternatives can be distinguished or not."

R2.13 The optical path length difference analyses in the Methods section are important considerations for indistinguishability of the emitted pairs from the three crystals. The authors clearly understand this.

We appreciate the Reviewer's encouraging feedback. This gives us more confidence to build our argument to rebut Reviewer #1, that the present nonlinear interferometer, composed of three alternative processes, is fundamentally different from the traditional linear interferometer and triggers new phenomena that can't be realised in the linear interferometer.

R2.14 The theory of uncertainty relations and wave-particle duality has progressed to much more descriptive and nuanced versions of the $D^2 + V^2 \leq 1$ relation, even for binary interferometers (see, e.g., Coles et al. "Equivalence of wave-particle duality to entropic uncertainty"). Therefore, it is hard to be satisfied by the interpretation of the "which path" information simply in terms of a definite downconversion source-- maybe there is a tradeoff between the visibility in this experiment and a variance or entropy of some measurement that can be considered as a more appropriate notion of "which path" information in the system and which does not lead to any interpretive contradictions. Can the authors provide any insight?

This is a very good point (or question). Thank the Reviewer for providing the information about the equivalence of the wave-particle duality relation (WPDR) and the entropic uncertainty relation (EUR). When we used the duality relation, we didn't know its equivalence to the entropic uncertainty relation. Following the Reviewer's remark, we have searched the literature. A recent experiment has also demonstrated the equivalence of EUR and WPDR [Daniel Spegel-Lexne et al., *Sci. Adv.* 10, eadr2007 (2024)]. The EUR was also reviewed in [P. J. Coles et al., *Rev. Mod. Phys.* 89, 015002 (2017)]. We note that the EUR has even been extended to multipath interferometers [P. J. Coles, *Phys. Rev. A* 93, 062111 (2016)]. We have carefully read these papers. No matter whether it's a two-path or a multi-path interferometer, the authors connect some form of entropy to the path distinguishability and fringe visibility through guessing games. They show that many WPDRs in the literature are special cases of a generic EUR. The entropic view provides an operational framework for analysing the duality relations. In the case of a two-path (binary) interferometer, the visibility is related to the max-entropy and the distinguishability is related to the min-entropy, which quantifies the lack of information about the particle behaviour.

For the two contradictory ways to ascribe the "which-path" information, it's enough to use the two-path WPDR/EUR because we have two sources equivalently. Note that the particle behaviour is quantified through the which-path observable $Z = \{|0\rangle\langle 0|, |1\rangle\langle 1|\}$, which corresponds to the upper and lower path in the Mach-Zehnder interferometer, in the entropic view. In the same spirit, we can also use this qubit basis to denote that the photons come from S1 (S1') and S2 (S2'). The wave observable can be defined through the phase applied between the two effective sources. Therefore, the same arguments can still be used to derive these two contradictory perspectives through this kind of WPDR/EUR.

If the system is treated as a whole, we believe the WPDR/EUR in the three-path interferometers can be applied in our case. As the paper [P. J. Coles, *Phys. Rev. A* 93, 062111 (2016)] has shown, the definition of distinguishability \mathcal{D} can be directly generated to n paths. The which-path observable is just associated with the path the photon takes inside the interferometer. Therefore, the notion of "which-path" information can still be defined through the entropy of the measurement of which source the photons come from in the present three-crystal experiment. It's only that the visibility \mathcal{V} needs to be redefined using the maximal guessing probability at the interferometer output. In this scenario, the which-path information can still be interpreted in terms of a definite source from which the photons originate. They may not lead to interpretive contradictions because the two quantities quantify the available particle and wave information of the system as a whole, but the duality relation still holds. We still cannot draw a definitive conclusion now. Besides, there is no universal formula for WPDRs with more than two paths [P. J. Coles et al., *Rev. Mod. Phys.* 89, 015002 (2017); B. G. Englert et al., *Int. J. Quantum. Inform.* 06, 129 (2008)], which means we need to adopt a proper definition which captures the underlying physics in our setup. In addition, our setup is a nonlinear interferometer with two detectors. The "waves" behaviour cannot be captured by the interference pattern, with extreme interference corresponding to only one detector clicking and no interference corresponding to a uniform distribution over all detectors. We think this may be a good future research topic. Through a proper definition of visibility and distinguishability in a three-path nonlinear interferometer, we can experimentally investigate the WPDR and the equivalence of WPDR and EUR in higher dimensions.

We have made slight additions to the discussion about the entropic uncertainty relations to clarify the Reviewer's concerns.

Line 624: "We note that some theoretical works have provided a general framework to derive wave-particle duality relations (WPDRs) based on entropic uncertainty relations (EURs) [25]. The equivalence of WPDR and EUR has also been demonstrated in a recent experiment for two-path interferometers [26]. In the entropic view, visibility and distinguishability can be defined through some kind of entropy, which quantifies the information of the particle and wave behaviour obtained in the system. The extended framework for formulating the WPDR from EUR in multi-path interferometers [27] provides experimental insights into investigating the WPDR and its equivalence to EUR in higher dimensions, based on our current setup. This may be the subject of our follow-up work."

Reviewer #3 (Remarks to the Author):

In this manuscript, the authors consider a frustrated SPDC system with three nonlinear crystals to investigate deducible “source information” when grouping the crystals in different ways. Theoretically, they have observed that: If (1) each crystal emits photon pairs of the same amplitude and (2) the relative phase between two sequential crystals in the photon’s path is set at π , the overall photon-pair generation rate of any two sequential crystals combined vanishes. Since we can meet these two conditions simultaneously for both the first & second crystals, as well as the second & third crystals, this appears to create a paradoxical situation where, even though the overall generation rate is nonzero, the photon pairs are seemingly not generated from any pair of the crystals. They use this final observation to conclude that “it is impossible to ascribe a definite physical origin of the photon pair even if the emission probability of one individual source is zero and full path information is available”. They have also performed carefully designed experiments to confirm their theoretical findings.

We thank the Reviewers for their thorough review of the manuscript and for their constructive comments on our results. This is a good summary of the present work. What follows is a point-by-point response to the issues raised by the Reviewer.

R3.1 While their observations and findings are thought-provoking, some gaps leading to their conclusion need further clarification. To start with, the whole reasoning seems to rest on the following premise in the abstract: “if visibility is zero, a high path distinguishability can be obtained”. However, if we look at the visibility-distinguishability tradeoff given in [4], we see that these quantities satisfy an inequality, rather than an equality. This means that, even in that context, the above premise does not necessarily hold. Although the tradeoff inequality can be saturated for certain pure states, to our knowledge, it has not been established that the inequality holds as an equality for all pure states. Even if the equality does hold for all pure states in an interferometric setup, it is unclear whether this quantitative expression relating visibility and which-way information can be translated into a similar tradeoff between visibility and “which-source” information in the non-interferometric setup considered in this work. Importantly, in the former scenario, the amplitudes and the relative phase are between the two different arms/ kets, which is not the case here. To this end, it is also unfortunate that notions like “path information” and “source information”, which are central to the whole argument, were not properly introduced. Consequently, one cannot even readily verify the validity of claims like “full path information is available”; or are we supposed to understand this to mean that the visibility is zero?

We thank the Reviewer for these thoughtful comments. We agree with the Reviewer that visibility zero does not automatically mean high path distinguishability. Nor do we claim to have a completely pure state in the experiment. However, our arguments do not rest on this premise. The reasoning can be broken down as follows:

- 1) We define a frame of viewing the interferometer by grouping two crystals together and arriving at a two-source interferometer.
- 2) We actually perform a which-path experiment in this two-source interferometer by successively measuring the count rates with S1 blocked and with S2 blocked. The high degree of “path distinguishability” is based on this measurement. This is because if S2 is blocked, a very low count rate can be observed (from S1), and if S1 is blocked, a high count rate is observed from S2. It is not a priori related to visibility.
- 3) We analyse the interferometric visibility and find that the observed low visibility is consistent with the high degree of path information, so that the duality relation is satisfied (see response to comment R1.8). The physical interpretation is that more than 90% of photon pairs originated from S2.
- 4) We do the same which-path measurements for S1' and S2' and find also that a high degree of path information is available. However, now a physical interpretation of this path information is that more than 90% of photon pairs originated from S1'.

The intermediate case, when $D^2 + V^2 < 1$, accounts for the partial distinguishability. However, one doesn't need the inequality to be saturated when one uses it in practice. We emphasise that, in the intermediate case, a **high probability** of obtaining path information can be achieved only with **low visibility**. The quantitative estimation of the quantity of $D^2 + V^2$ (~ 0.91 for S1&S2, ~ 0.93 for S1'S2') is given in the response to Reviewer #1's comment R1.8 (see Supplementary Table A1 in R1.8). We see that they still have a trade-off, which means one can win with a high probability (although not 100%) to distinguish where the photons come from.

Regarding the second point, first, our setup is an interferometric setup. This is a nonlinear interferometer which is based on the SU(1,1) transformation (see the original work by [Yurke et al., Phys. Rev. A 33, 4033 (1986)] and also the frustrated-down conversion by [Herzog et al., Phys. Rev. Lett. 72, 629 (1994)]). Different from the setup by Herzog et al. and other interferometers, the Reviewer probably think our setup only has one arm; how can it be an interferometer? If the Reviewer refers to the comments of Reviewer #1 (e.g., R1.11, R1.12. Here we also kindly invite the Reviewer to R1.11 and R1.12) and our responses, they will know that the essence of the interference is actually the indistinguishability of the different alternatives. In response to comment R1.11, we present three process alternatives which contribute to the three indistinguishable probability amplitudes. The amplitudes and relative phase are between these alternatives. From this perspective, they construct an interferometric setup. The only difference is that, in our case, signal and idler photons are collinear. However, this is feasible, as shown in Fig. 3(a) with two crystals in [Chekhova and Ou, Adv. Opt. Photon. 8, 104 (2016)], it's just a different geometry. For convenience, we also present the figure below. One can also think of it from another perspective. Although the signal and idler are collinear, they have different polarisations. Thus, their phases are changing differently. Also, the pump has a different wavelength. Therefore, when one rotates the phase plate, the relative phase does change. This can produce the interference fringe. This has the same effect as changing the relative phase of the pump, signal, and idler in Herzog's and other similar interferometric setups.

[editorial note: figure redacted]

Fig. 3 from [Chekhova and Ou (2016)]

If the Reviewer understands our setup is an interferometric setup, then the second point will be easy to clarify. For the frustrated down-conversion with two sources, the photon pair emission rate, $R \propto |a + e^{i\varphi}b| \propto a^2 + b^2 + 2ab \cos \varphi$, where a and $e^{i\varphi}b$ correspond to photon pair emissions from the first and the second source, is described by the interference law analogous to the description of a particle in a two-path interferometer. When translating the duality relation into our setup between the visibility and "which-source" information, the visibility V is still the visibility of the interference fringe. It's obtained using the minimum and maximum count rates in a way analogous to traditional interferometry.

$$V = \frac{I_{max} - I_{min}}{I_{max} + I_{min}} = \frac{2ab}{a^2 + b^2}$$

It quantifies the amount of suppression and enhancement of the photon pair emission rate. The path distinguishability D in this case corresponds to the distinguishability of "which-source" produced a photon, and can be quantified in a similar way using amplitudes a and b as in the traditional two-path interferometer.

$$D = \frac{a^2 - b^2}{a^2 + b^2}$$

Therefore, the quantitative expression relating visibility and “which-way” information in a traditional two-path interferometer can be translated directly into the trade-off between visibility and “which-source” information in the present work.

To clarify the notions of “path information” and “which-source information” and also conform to Reviewer #2’s comment R2.7 to be consistent with the term, we have added the following text to the manuscript using the notions of “which-path information” and “which-source information”:

Line 1773: “Here we show that the duality relation can be applied directly to our setup. For interferometry with two crystals, the photon pair emission rate is described by an interference law analogous to the two-path interferometer, i.e., $R \propto |a + e^{i\phi}b| \propto a^2 + b^2 + 2ab \cos \phi$, where a and $e^{i\phi}b$ are the corresponding probability amplitudes contributed by these two crystals, and ϕ is the relative phase between these amplitudes. To apply the duality relation to our setup, we define the interference visibility V similar to the traditional two-path interferometer,

$$V = \frac{R_{max} - R_{min}}{R_{max} + R_{min}} = \frac{2ab}{a^2 + b^2}$$

The distinguishability D in this case corresponds to the distinguishability of “which-source” produced a photon pair, and can be quantified in a similar way as in the traditional two-path interferometer [2],

$$D = \frac{a^2 - b^2}{a^2 + b^2}$$

With this, the duality relation can be directly used in our setup. The “which-source” information is simply that the photon pair is produced at which source and travels all the way to the detector. When full path information is available, this means one knows where the photon pair originates.”

R3.2 Coming back to the observations, if we think of NL1 & NL2 as one source, the authors argued that their combined probability amplitude is zero. While this statement looks innocent, does this mean we are entitled to think of

“combined probability amplitude of A and B is zero = the *emission* probability of both A and B as zero”?

This is not our claim. There is no place in the manuscript where we mean “combined probability amplitude of A and B is zero = the *emission* probability of both A and B as zero”. It simply means that the combined emission probability of A and B is zero, or no photons were emitted from the combination of A and B. In fact, the individual emission probability of A=NL1 and B=NL2 is not zero. But this doesn’t influence our reasoning. We only care whether the combined single-source of A and B emits photons. If the combined source emits no photons, then one concludes that the photons come from another source C=NL3.

From Equation (1), there seems to be no legitimate reason to consider only the probability from summing the amplitude of two of the three terms, all associated with the *same* ket vector $|s\rangle|i\rangle$. Even if the above statement is valid, when A = NL1 & B = NL2, one must conclude that NL3 must have contributed to generating the photon pairs in this case. Similarly, if A = NL2 & B = NL3, one must conclude that the photon pairs are generated by NL1. This seems to be the basis for the authors’ claim that we now have two “contradictory which-way information”, when both A = NL1 & B = NL2, and A = NL2 & B = NL3 conditions are satisfied. However, at least three arguments tell us that such reasoning might not be proper, or must be trodden on carefully.

This comment is somehow related to the comments R1.11 and R1.17 of Reviewer #1. This is true, yet this is done in the mathematical description of virtually EVERY experiment as the physical processes that could happen in principle are mapped to probability amplitudes. The emission from a SPDC crystal, or from a slit or a light beam is very commonly associated with a single ket vector. Yet another experimenter might consider the substructure of their experimental system and assign, for instance,

different ket vectors to different slices of the crystal.

The essential claim of our work is that there is some freedom to choose a mapping between physical processes and the quantum formalism, and we must be very careful when drawing conclusions from the quantum formalism about physical reality. For example, path information might have a different physical interpretation depending on how one maps kets to physical processes. There exist both **mathematical** and **physical** meanings of how one looks at the probability amplitude and the probability. As we presented in the response to comment R1.11, mathematically, one is inclined to sum the amplitudes of two terms. For example, if $|A + B|^2 = 0$, one can say $A + B = 0$ in $|A + B + C|^2$ and think $P = |A + B + C|^2 = |C|^2$, or the other way, if $|B + C|^2 = 0$, one can say $B + C = 0$ in $|A + B + C|^2$ and think $P = |A + B + C|^2 = |A|^2$.

When considering the system as a whole, one should take into account the contribution of all three terms to the total probability, as they are associated with the same ket vector, as the Reviewer stated. However, this interpretation is based on the experimenter's **observation** that three crystals are presented in the setup. As shown in the response to comment R1.17, we present two Gedankenexperiments, where one can interpret the experiment from two perspectives if the three crystals are placed in two black boxes in different ways. This is the essence of the present work: the interpretation of path information in the experiment is subjective. Different experimenters may reach different conclusions depending on how they conceptualise the experiment. These two perspectives correspond to the two mathematical meanings above. Therefore, it is not about whether the reason is legitimate or not to consider only summing the two probability amplitudes; it is about how the experimenter/observer interprets the experiment based on their different assignments of probability amplitudes to physical processes. It is simply that they utilise these two mathematical meanings, which sum the two amplitudes, when interpreting their observations.

First, if we push this view further and think that interaction between the fields of the signal/ idlers from NL1 & NL2 happens before the interaction with NL3—after all, the laser field first passes NL1 before NL2, and finally NL3—then one may argue that the NL1 & NL2 combined emission probability amplitude must be zero first, while the non-zero probability emerges later after the laser hits NL. So, whatever is detected must be emitted from NL3. Then, visibility is zero, source knowledge = NL3, and so there's no ambiguity of "which-source" knowledge; everything appears consistent.

We thank the Reviewer for this intriguing argument. Fortunately, we also considered this problem when conceiving this experiment. This somehow provokes people to relate it to relativity, considering the timing order of the interaction. There are two aspects that we can argue about in this regard.

First, the propagation time of the laser is only relevant during the first few nanoseconds when the laser is on. However, this time duration is so short. After the laser had passed through all the crystals, the detectors detected the photons. This means that all crystals are already in action when the laser is stable, and the pair interference occurs simultaneously. Image NL1, NL2, and NL3 produces signal/idler photons at t (They can produce at the same time as we have argued the laser has arrived at them. It is only the laser in the same time-bin that arrives at them differently. But the laser in different time-bins can interact with crystals at the same time. One should be careful about this). Then the signal/idler photons from NL1 propagate to NL2 after Δt and interference begins to happen. But it also takes the same time Δt for photons from NL2 to NL3 and begins to interfere. Therefore, in fact, these two interference processes happen simultaneously.

Second, from the experimenter's perspectives, both NL1&NL2 and NL2&NL3 are valid. As we already argued above, the two Gedankenexperiments actually represent these two perspectives. The experimenter cannot determine whether NL1&NL2 or NL2&NL3 occurs first when observing only two black boxes and interpreting the results based on their observations (we kindly invite the Reviewer to our response to comment R1.17).

Besides, recall the comments R2.12 by Reviewer #2: "the description of a system of quantum emitters (**even if it is composed of spatially distinct parts in a laboratory**) must encompass all the possible emission outcomes according to the sum of corresponding amplitudes". All alternatives contribute to the

interference; no interference happened before the other.

Second, what prevents the possibility that *some* of the photon pairs are from NL1 & NL2, and some from NL2 & NL3? This point of view is consistent with the experimental results as well.

We agree that it is possible that some of the photon pairs are from NL1&NL2 and some from NL2&NL3 in the experiment. However, this is caused by the experimental imperfection. In an experiment, it is not possible to have zero photons from NL1&NL2 if the phase between them is π and NL2&NL3 are the same. Therefore, we see few photons from NL1&NL2 or NL2&NL3 at the destructive point π . However, one still has a high probability of knowing the photon's origin, as is shown in the probability $p_3 = 95.14 \pm 0.59\%$ and $p_1 = 96.41 \pm 0.47\%$ in the main text, if one conceptualizes the experiment as the above-described Gedankenexperiments.

On the other hand, we understand that the Reviewer also means the theoretical possibility that some photon pairs are from NL1&NL2 and some from NL2&NL3. We admit there may exist this possibility. But this does not weaken our claim. If the Reviewer can identify a realisable situation, such as the above two Gedankenexperiments, in which the experimenter conceptualizes it in this way, then this is another **subjective interpretation** that strengthens our argument. However, we believe this possibility is equivalent to when the black boxes are opened and all three crystals are presented, as described in the third viewpoint in the above response to comment R1.17. In this situation, it is not possible to determine where the photons originate, or one can simply state that the photon pairs come from the system as a whole.

Third, recall Feynman's important lesson about quantum theory: we have to add all the relevant amplitudes before computing the probability. So, it may simply be incorrect to think of zero emission probability of both NL1 & NL2 (or NL2 & NL3), as this possibility happens, via Equation (1), in superposition with the emission from NL3 (or NL1).

We thank the Reviewer for this comment. This is basically the same issue raised by Reviewer #1 in comment R1.11. One should be cautious when looking at this problem. Through all the arguments, we are utilising Equation (1), which accounts for all the relevant amplitudes. Nothing contradicts Feynman's lesson. It's only in the special case, where $\phi_A = \phi_C = \pi$ and $|A|^2 = |B|^2 = |C|^2$ in $P = |A + B + C|^2$ ($A = ae^{i\phi_A}, B = b, C = ce^{i\phi_C}$), one has either $A + B = 0$ or $B + C = 0$. In this case, one still adds the relevant amplitudes. It is merely one of the probability amplitude terms that is zero ($P = |0 + C|^2$ or $P = |A + 0|^2$). But they are still in superposition. Note, however, the individual terms $A, B,$ and C are not zero, they are still contributing to the total probability. The two mathematical forms ($P = |0 + C|^2$ or $P = |A + 0|^2$) represent the two subjective interpretations one has in the above two Gedankenexperiments.

As argued above, we never claim that the emission probability of both A and B is zero when their combined probability amplitude is zero. Here, the Reviewer is still using the first inference "combined probability amplitude of A and B is zero = the *emission* probability of both A and B as zero" from above to refute our argument. There is a possibility that photon emission from NL1 and NL2 occurs individually. However, when the combined probability amplitude is zero, e.g., $A + B = 0$, it's enough for one to use this fact to try to interpret the experiment and get the conclusion that the photon pairs come from C=NL3, as demonstrated by the Gedankenexperiments. It is merely that the experimenter interprets the experiment solely on the basis of their observations and understanding, which corresponds to the above two mathematical forms while still obeying the quantum rules.

Even considering it simply, one only has two sources, NL1 and NL2, which emit with probability amplitudes A and B . Assume NL1 has zero efficiency to produce no photons, then $A = 0$. But one still calculates the total probability as $|0 + B|^2$. Nothing violates the superposition rules.

R3.3 All these "inconsistencies" point to another potential pitfall in concluding from quantum mechanical predictions using classical "intuitions". Presumably, this is the basis for the following passage in the abstract "our findings shed new light on the physical interpretation of probability assignment and path

information beyond its mathematical meaning and reshape our understanding of the whole and part in the context of distinguishability.” However, is it not also evident from the first paragraph of the Discussion and conclusion that the lesson to be learned here is, to some extent, already known in [8] (published more than 30 years ago) and [19] (in the PhD thesis of one of authors, published more than 4 years ago). If so, in what way has this shed new light on our understanding?

We thank the Reviewer for this remark. As argued above, there are no inner “inconsistencies” in our reasoning, only some unclearness. We have added some sentences in the revised manuscript to clarify the Reviewer’s concerns.

The lesson to be learned here is very different from the frustrated down-conversion with two crystals in [8]. With two crystals, one can win with high probability to get a “which-source” information if one of the two sources emits photon pairs with a higher probability. In this sense, the physical interpretation and the mathematical meaning are consistent, as illustrated in the response to comment R1.11. However, in the three-crystal case, the physical interpretation and mathematical meaning are discrepant. There are two mathematical possibilities that one can use to conceptualize the experiment. These conceptualizations show that the “which-source” information one obtains is very subjective. It depends on how one looks at the experimental setup. Even more deeply, it shows that what one conceptualizes is just what the universe wants to show you. However, this may not reflect the truth of the universe, as our perceptions are based solely on our five senses (we rely on our senses to interpret experimental data). However, this falls outside our scope, as we are discussing the physical implications.

Another possible lesson to be learned is how the whole and part should be defined in the context of distinguishability. In the three-crystal setup, one sees that each crystal contributes an interference term/alternative, which is added together to get the total probability. These terms/alternatives are indistinguishable from each other.

As for the PhD thesis, scholars are permitted to publish research articles in addition to the thesis. This is based on the theoretical proposal in the thesis. We finished the experiment, obtained and analysed the data, which are not included in the thesis, to demonstrate the theory. Therefore, we believe the work in the thesis of one of the authors to be published in a journal is permitted and should not be a reason to refute our publication just because the proposal already appeared in the thesis. If so, many publications should be withdrawn simply because they contain the same works in their theses.

To summarize, while we agree that this work is thought-provoking and can potentially help us appreciate better the role of interference in a “which-source” setup, its current presentation does not make a strong case for publication in Nature Communications. Besides the issues raised above, the authors can also help improve the clarity of the work by addressing the following issues:

We have added some notes in the Supplementary Information to clarify the concerns raised by both Reviewers #1 and #3 above.

Line 1634:

“Supplementary Note 1: Interference of three processes

In this section, we show that our setup is a nonlinear interferometer with three processes and the interpretation of path information is indeed subjective. The nonlinear interferometer incorporates a $SU(1, 1)$ transformation and is fundamentally different from the traditional $SU(2)$ Mach-Zehnder and Hong-Ou-Mandel-type interferometer [29]. It describes an *active* optical dynamics, which means that the ports of the interferometer can be left in vacuum. Therefore, one can treat our setup as three alternative processes that occur within each crystal as a result of second-harmonic generation. In Supplementary Fig. A1, we plot the three processes. The first process I is from the pump laser to crystal NL1 and to the detector. The second process II is from the pump laser to the crystal NL2 and to the detector. The third process III is from the pump laser to the crystal NL3 and to the detector. Each crystal contributes to an alternative, that is, NL1 to $A = ae^{i\phi_A}|s\rangle|i\rangle$, NL2 to $B = b|s\rangle|i\rangle$, and NL3 to $C = ce^{i\phi_C}|s\rangle|i\rangle$. When treating the system as a whole, one accounts for all these terms and interprets the probability as $P_{three} = |A + B + C|^2$.

In the two-crystal case (NL1 contributes to $A = ae^{i\phi_A}|s\rangle|i\rangle$, NL2 contributes to $B = b|s\rangle|i\rangle$), one has $P_{two} = |A + B|^2$. Assuming $A = 0$, we have $P_{two} = |0 + B|^2 = CC$. An experimenter is asked to determine the origin of the detected photons. The experimenter can perform the experiment. They first remove NL1 or filter the SPDC photons from NL1 and find that the detectors have counts CC . Then, they move NL1 in or remove the filter and find that the counts are still the same. The experimenter also operates NL2 in the same way. They will find that the counts are zero when NL2 is removed and that the counts remain CC when NL2 is present. Thus, the experimenter concludes that the photons originate from NL2. This is in accordance with the mathematical meaning. Therefore, one can have a high probability of knowing where the photons come from if one source emits no photons in the two-crystal case. The mathematical meaning and the final physical interpretation, based on the observation of the experimenter, are consistent.

In the three-crystal case, different experimenters can arrive at different conclusions for the same underlying setup. Conceiving the Gedankenexperiment (see Supplementary Fig. A2): Two experimenters (E1 and E2) receive a setup composed of two black boxes (BB1, BB2), and a phase shifter is inserted between these two boxes. For E1, NL1&NL2 is placed into BB1 and NL3 is in BB2. No photons are emitted from NL1&NL2. For E2, NL1 is placed into BB1 and NL2&NL3 is in BB2. No photons are emitted from NL2&NL3. Similarly, the experimenters are asked to determine the origin of the photons. Based on their experience in the two-crystal case, the experimenters can perform a similar operation by treating each black box as a source. First, they remove BB1 and then reinsert it; second, they remove BB2 and then reinsert it. After this, experimenter E1 comes to the conclusion that the photons come from BB2=NL3 based on their observation, whereas experimenter E2 arrives at the conclusion that the photons come from BB1=NL1 based on their observation. One sees that fundamentally both experimenters have the same experimental setup. The different conclusions they obtained are simply because they conceptualize the experiment differently, or the experimental setup manifests itself differently. These two perspectives just correspond to the two different mathematical possibilities, i.e., $P_{three}^{(1)} = |0 + C|^2$ and $P_{three}^{(2)} = |A + 0|^2$, if $\phi_A = \phi_C = \pi$ and $|A|^2 = |B|^2 = |C|^2$. Here, one can see that interpretations of path information are highly subjective. It depends on how one conceptualizes the experiment's configuration. Furthermore, if a third experimenter finds that one black box has two crystals and opens the black boxes, they will interpret that all three crystals contribute to the counts if they are quantum physicists and believe in quantum rules. Or, in an ingenious way, they can say that the photons come from the system as a whole. In the case of two crystals, there is no inconsistency in the interpretation of the path information. The inconsistency and subjectivity in the interpretation of path information emerge only in the three-crystal case.”

R3.4 Lines 323/343: The authors should explain clearly what constitutes a 4f system. The descriptions around Line 323 and those around Line 343 are not consistent. Is it defined by lenses between each pair of crystals (as described in the former), or is it a pair of crystals (as described in the latter)?

We thank the Reviewer for pointing out this inconsistency. The 4f system is defined by lenses between each pair of crystals. To correct this, we revised the sentence:

Line 344: “The lenses between each pair of crystals form a 4f system to obtain good spatial overlap of the SPDC photons from each crystal.”

Line 385: “Each crystal is enclosed by two lenses, and two lenses between each pair of crystals form a 4f system.”

R3.5 Lines 399 & 401: “canned” should have been “scanned”

As noted in R1.7 of Reviewer #1, we have corrected the same errors.

R3.6 Line 417, 741-742, 745, 762: The hyperlinks for Extended Data Fig. 1b, Extended Data Fig. 1, Extended Data Fig. 2, and Extended Data Fig. 3 do not link the Figures under Extended Data, but to

the main text.

Thank the Reviewer for their eye for detail. We have linked to the corrected figures in the revised manuscript.

R3.7 Lines 497-498: From the paragraph after Eq. (4), it seems like the off (on) status corresponds to one setting the amplitude to be equal in magnitude and the relative phase to π (or not). However, from the description given here, the phases are both set to π . Then, what distinguishes the on/ off status of the crystals NL1 or NL3?

We thank the Reviewer for spotting this. Some confusions are also pointed out in R1.9 by Reviewer #1. The on/off status of the crystals is not determined by the settings. The on status is when the crystal is active in the optical path and generates SPDC photons. The off status is when the crystal is removed from the path or its SPDC photons are filtered. Therefore, when we say the crystal NL3/NL1 is off, it means that NL3/NL1 is removed from the path, or a short-pass filter is used to filter the SPDC photons. With this, when NL3 is off, only NL1 and NL2 generate SPDC photons. When the relative phase ϕ_A between NL1 and NL2 is set to π , ideally, no photons come from them, i.e. $CC_{S1} = 0$ ideally. However, to account for the imperfection in the experiment, we use the CC_{S1} (there are still a few counts) when NL3 is off to **estimate the counts CC_{NL3} when all crystals are on**. Further, we use the total counts CC_{tot} when all crystals are on to minus this small quantity CC_{S1} . Through this, we can estimate the probability that the photons come from NL1 and NL3 in these two perspectives.

Please refer to the response to R1.9 for the revised sentences.

R3.8 Lines 485-505: Following the last comment, the coincidence count rates $CC_{\{NL3\}}$ and $CC_{\{NL1\}}$ were presumably carried out under different experimental settings (with different crystals turned off). If so, what entitles us to compare and even combine the probability of photon pairs generated in the individual crystals? And why should one expect that this has anything to do with the situation when all crystals are turned on?

That is correct, the coincidence counts have been measured by placing filters in the setup that allow us to block the emission from individual crystals. However, the counts of NL1&NL2 and NL2&NL3 are only used to estimate the minor quantity CC_{S1} and $CC_{S2'}$ in the experiment because we are not in the ideal case. To be specific, CC_{S1} is estimated when NL3 is blocked (off) and $CC_{S2'}$ is estimated when NL1 is blocked (off). These quantities can be treated as errors that deviate from the ideal case (ideally, they should be zero). However, when calculating the probabilities p_1 and p_3 , all crystals are already turned on. Thus, we can use the total counts CC_{tot} when all crystals are on to minus CC_{S1} and $CC_{S2'}$ to get the probabilities p_3 and p_1 , respectively. This entitles us to combine and compare the probability of photon pairs generated in crystal NL1 and NL3. It is analogous to a which-path measurement in a standard interferometer. When drawing conclusions from it, the changed configuration of course needs to be taken into account, which has been analysed in all duality relation papers [Greenberger et al., Phys. Lett. A 128, (1988); Herzog et al. Phys. Rev. Lett. 75, 3034 (1995); Jaeger et al., Phys. Rev. A 51, 54 (1995); Englert, B. G., Phys. Rev. Lett. 77, 2154 (1996)]. We do this by characterising path information and visibility according to the definitions of [Greenberger et al., Phys. Lett. A 128, 391 (1988)].

R3.9 Line 1048: Did the authors mean to write $\phi_A = \pi$ or $\phi_C = \pi$?

Yes, we have rewritten it to avoid confusion.

Line 1168: "At the point $\phi_A = \pi$ or $\phi_C = \pi$, one can see that interference visibility is zero."

R3.10 Line 1072: How are the counts of Extended Data Fig. 2 normalized for different choices of ϕ_A ? And why are the curves for $\phi_A=0$ and $\phi_A=2\pi$ different?

Theoretically, if $a = b = c$, then counts $R(\phi_A, \phi_C) \propto |e^{i\phi_A} + 1 + e^{i\phi_C}|^2$. The normalization is done by dividing $R(\phi_A, \phi_C)$ with the maximum value 9, i.e., $R(\phi_A, \phi_C) = |e^{i\phi_A} + 1 + e^{i\phi_C}|^2 / 9$. All the different

choices of ϕ_A are calculated this way. For each fixed ϕ_A , $R(\phi_A, \phi_C)$ can be plotted versus ϕ_C .

The curves for $\phi_A = 0$ and $\phi_A = 2\pi$ are the same. They overlap. Therefore, the blue curve for $\phi_A = 0$ can't be seen.

R3.11 Line 1214: It would be helpful to indicate either on the images or in the caption which images are for the crystal plane and which others are for the Fourier plane.

Thank you. We have indicated in the caption of Extended Data Fig. 5.

Line 1353: "a-c are for the Fourier plane. d-f are for the crystal plane."

Reviewer #4 (Remarks to the Author):

We thank the Reviewer for their time and effort to review our manuscript.

Supplementary Information to comment R1.12:

We agree with the Reviewer that the origin of interference lies in the indistinguishability of alternatives. However, there are more points that need to be taken into account when considering what causes the indistinguishability of alternatives. To avoid unnecessary discussions in the main response, we outline some of our key viewpoints here.

For single-particle interference (such as a one-photon Mach-Zehnder interferometer), interference occurs when multiple path alternatives or processes leading to the same final outcome are indistinguishable. It's not about the particles being identical, but about the indistinguishability of the quantum alternatives (paths, histories, processes). However, things become subtle when dealing with multi-particle interferences (like in Hong-Ou-Mandel or boson sampling). We must be careful to consider what causes the indistinguishability/distinguishability of the alternatives. In such cases, particle identity enables the indistinguishability of alternatives. The two alternatives of the two-photon HOM effects become indistinct only if the photons themselves are identical. If two photons are not identical, the interference can disappear in principle.

In the strict quantum mechanical sense, particles are **identical** if they share the same **intrinsic properties**, such as mass, charge, spin, and all other internal quantum numbers. However, polarisation is not an intrinsic property of the photon; it's part of its quantum state. Otherwise, how can one say a photon is a boson based on the wavefunction symmetry of identical particles? Similarly, frequency or wavelength is also not an intrinsic property of the photon. It depends on the photon's energy, which is part of its quantum state. The arrival time is also an extrinsic property. However, when one discusses interference, one implicitly assumes the presence of a single type of particle or identical particles. Otherwise, why can't we observe Hong-Ou-Mandel interference between a photon and an electron? To our knowledge, this kind of Hong-Ou-Mandel interference between a photon and an electron has not been demonstrated. Therefore, identity is the premise of the interference in this sense.

Building on the above point, it is essential to distinguish between the definition of **identity** and **indistinguishability**. In brief, identity is about the type of particle (intrinsic properties), while indistinguishability is about the total quantum state (extrinsic properties). Only if we have identical particles can we continue to discuss the indistinguishability of the alternatives. In this sense, interference happens if states are indistinguishable. In the two-photon interference with non-degenerate entangled photons, the interference happens because the detectors are broadband (can't resolve frequency) or one post-selects on coincidences that erase frequency information. Therefore, the photons can still be indistinguishable at the level of joint detection and thus interfere with each other. However, if the detectors can resolve the frequency, then the distinguishability is restored, and the two-photon interference will disappear or be reduced. Some papers have shown this, for example, the theoretical work by Legero et al. (2003) "Time-resolved two-photon quantum interference" and the experiment by Kuo et al. (2016) "Spectral correlation and interference in non-degenerate photon pairs at telecom wavelengths". Kuo et al. generated non-degenerate photon pairs with polarisation-frequency entanglement and used a fiber spectrometer to record joint spectral intensity. They show that frequency distinguishability is enough to eliminate spectral interference (absence of fringes in Fig. 2(a) and (b) in the paper). The interference returns only when both frequency and polarisation information are erased. Another experiment, by Jin et al. (2015) "Spectrally resolved Hong-Ou-Mandel interference between independent photon sources", used a fast fiber spectrometer to show that when photons are frequency-mismatched, the traditional HOM dip was shallower with a reduced visibility. From this perspective, one can only say the two-photon interference effects were observed with two-photon systems comprised of **distinguishable** (not "non-identical" as the Reviewer said) photons if the distinguishable information of photons is erased or not resolved by the detectors.

The comments R2.14 of Reviewer #2 also provide us with some inspiration. From an information-theoretic perspective, any distinguishable information (intrinsic or extrinsic properties) printed on the quantum particles can be used to reveal the particle behaviour (through the interaction with the environment) as quantified by the entropy of a which-path observable [P. J. Coles et al., Nature Commun. 5, 5814 (2014)], thus smearing out interference because of the distinguishability of alternatives.

Regarding the paper mentioned by the Reviewer, "Can two-photon interference be considered the interference of two photons?", we have carefully read it. Yes, the photons do not appear at the beam splitter simultaneously. However, the "compensator" after the beam splitter ensures that the total travel time difference from the crystal to the detectors for the two-photon amplitudes becomes indistinguishable again. As a result, the two-photon amplitudes (r-r and t-t) become indistinguishable again at the detectors because each detector sees the photons arrive with the same overall relative timing difference. This experiment is effectively a quantum eraser scenario. The distinguishability of photons here refers to the case in which one considers each photon's individual path and arrival time before compensation or hypothetical direct measurement at the beam splitter. However, this distinguishability is removed before actual detection by the compensator. In Fig. 2 of the paper, labelling the time through the optical path travelled by a and b to be a_r , b_r and a_t , b_t for r-r and t-t cases, respectively. Thus, one gets $a_r - b_r = b_t - a_t$ (time interval between the two detectors is equal for the two cases) or $b_t - a_r = a_t - b_r$ (time interval of detecting the transmitted and reflected photons is equal for the two detectors), which indicates $a_r + a_t = b_r + b_t$. Therefore, the overall effective optical paths by a and b photons are still the same. One never measures each photon independently at the beam splitter; instead, one measures coincidences after the full apparatus (beam splitter + compensator). The measurement is set up precisely to hide any residual timing information. Hence, while single-photon paths are distinguishable in principle, the experimental design explicitly prevents access to this distinguishability in practice.

[editorial note: figure redacted]

In summary, the origin of the interference is the indistinguishability of the alternatives of the system as a whole. If the distinct properties (such as frequency, polarisation, arrival time, etc) of photons do not induce distinguishable information for the different alternatives, then interference occurs. This is the case when the detectors are blind to these distinct properties, or post-selection is done to smear out this distinguishable information. However, the indistinguishability of the photons themselves also counts when the involved properties of the photons matter. The meaning of indistinguishability in this context is the **inability** (detector or interferometer setup's resolution ability) to distinguish between quantum amplitudes of different alternatives, which contribute to the final detection when treating the system as a whole. If one builds an interferometer setup or uses a detector which can resolve the different alternatives by utilising these photons' distinct properties, then the indistinguishability of photons does count. The identity of the quantum particles is the basis for realising this indistinguishability.

Manuscript Title: Subjective nature of path information in quantum mechanics

REVIEWER COMMENTS & AUTHOR RESPONSES

Author Responses to Review Report – Round 2:

Reviewer #1 (Remarks to the Author):

I have carefully read the response letters submitted by the authors. I thank the authors for making such a comprehensive effort at addressing each concern/suggestion that I had raised in the previous round of review. In particular, I found many of their explanations quite insightful and thought-provoking.

The authors have resolved most of the concerns and suggestions that I had raised in the first round of review. In what follows, I would just like to outline a few residual concerns from my side. I mark the authors' comments from their response letter in blue normal font, *excerpt from the manuscript in blue italics*, and my own remarks in black normal font.

Firstly, I am still not convinced that the phenomenon reported in this work is peculiar to nonlinear interferometers and has no analog in linear one-photon interferometers.

- **R1.1:** The authors state that “Note that the meaning of path information in our three-crystal interferometer and the three-path linear Mach-Zehnder interferometer (MZI) is very different. In the MZI, the path information refers to the path the photon passes through. In the three-crystal interferometer, the path information refers to the place where the photon is generated. In the one-photon three-path linear MZI, the one photon must pass through all three paths, even though one blocks C and then blocks A&B when trying to regard A&B as path S1 and C as another path S2 (see Fig.~3). Because there is only one photon, one-third of the probability that the photon passes through each path...One cannot say this photon only passes through C and not through A and B; otherwise, how can one prove that A and B have a destructive interference if it only passes through C? However, with the three-crystal case, one can conclude that the detected photon pairs originate from crystal C=NL3 because it is reasonable that the detected photons are only generated by C and travel to the detectors when A and B are blocked.”

I am not sure I agree with this reasoning. In both the three-crystal case and the linear MZI case, one can imagine a destructive interference between A and B. Alternatively, one can also imagine a destructive interference between B and C.

In this regard, I think it is important to not get confused by the parallel geometry of the linear SU(2) Mach-Zehnder interferometer and the series geometry of the nonlinear SU(1,1) interferometer. Although the setup geometries are different, as far as the superposition of probability amplitudes corresponding to different alternatives is concerned, both situations are perfectly analogous.

Response: We thank the reviewer for this thoughtful comment and for pointing out the close analogy between the linear SU(2) Mach-Zehnder interferometer (MZI) and the nonlinear SU(1,1) interferometer at the level of probability amplitudes. We agree that both systems can be described by a coherent superposition of alternative probability amplitudes, and that destructive interference can, in principle, be assigned to different pairs of alternatives depending on the chosen decomposition.

Our intention in the quoted paragraph was not to deny this analogy, but rather to clarify a **conceptual distinction in the physical interpretation of “path information”** in the two scenarios.

In a linear three-path MZI with a single photon, the different paths correspond to mutually exclusive propagation modes of the same photon. Even if one path is physically blocked, the interpretation of interference necessarily relies on the superposition of propagation amplitudes of that photon through the remaining open paths. In this sense, assigning the detected photon to a single path while excluding the others is not meaningful because the observed interference (e.g., destructive interference between A and B) fundamentally requires the coexistence of those amplitudes.

By contrast, in the three-crystal $SU(1,1)$ interferometer, the alternatives correspond to **distinct photon-pair generation events occurring in spatially separated nonlinear crystals**. When alternatives A and B interfere destructively, the existence of the photon pair itself is subject to the interference. Here, the “path information” refers to the origin of photon-pair creation, rather than to a propagation path of a pre-existing photon.

We fully agree with the reviewer that, at the level of probability amplitudes, one may equivalently describe interference as occurring between any pair of alternatives (e.g., A–B or B–C), and that the parallel versus series geometry alone does not invalidate the analogy. To avoid possible confusion, we will revise the manuscript to make it clearer that our claim concerns the **physical interpretation of which-path information**, not the mathematical structure of interference. We will emphasize that, while the interference mechanisms are formally analogous, the ontological meanings of the alternatives (propagation paths versus generation processes) differ in the two systems.

We thank the reviewer for helping us improve the clarity and precision of this discussion.

- **R1.2:** The authors then state that “Thus, we can interpret the traditional MZI in two dual ways: (1) Path interference (path histories or path alternatives, as often mentioned by the Reviewer): A photon can take two paths, and the outputs result from the quantum superposition of these paths, which is a particle picture. (2) Field interference (as mentioned by the Reviewer’s comment R1.19): Optical fields split and recombine, and their amplitudes interfere, which is a wave picture. However, in the nonlinear interferometer, the interference occurs between “the vacuum evolving through the second crystal” and “the amplified field coming from the first crystal” [Yurke et al., Phys. Rev. A 33, 4033 (1986)]. In this sense, it’s not just an interference of different path alternatives, but an interference between two processes, i.e., coherent superpositions of different pair-generation amplitudes. Therefore, we can refer to it as the interference between probability amplitudes corresponding to process alternatives.”

I would just still argue that on a fundamental level, there is no difference between interference of different path alternatives and the interference of difference process alternatives. This is because **the notion of “paths” in quantum theory** does not necessarily refer to paths in real position space, but **in general refers to “paths” in an abstract configuration space**. In the linear MZI, these paths just happen to correspond to actual paths that a photon can take in real space, whereas in the 3-crystal nonlinear interferometer, these “paths” correspond to ways in which the pump photon can get destroyed to produce a two-photon state. But in the end, it is still the same principle of linear superposition in action and results in interference between amplitudes corresponding to these paths.

Also, regarding the statement that in a nonlinear interferometer “the interference occurs between “the vacuum evolving through the second crystal” and “the amplified field coming from the first crystal” [Yurke et al., Phys. Rev. A 33, 4033 (1986)], I am not denying this interpretation but one could equally well argue that the interference occurs between probability amplitudes corresponding to the different alternatives by which down-conversion can occur. For instance, please see Appendix A of the paper P. Sharapova et al. Phys. Rev. A **91**, 043816 (2015). The equation A1 has been obtained by linearly superposing the probability amplitudes for down-conversion happening in crystal 1 and crystal 2.

I also looked at the reference cited here:

“This process interference was even demonstrated in the experiment, when the two processes are distantly separated [Pseiner et al., Phys. Rev. Res. 6, 013294 (2024)]. Note that this is not two-photon interference, despite having two photons.”

I do not see how this refutes my observation. However, if the authors have only cited this reference to argue that this is interference between processes and not paths in real space, then I agree with them. But as I have pointed out previously, interference in quantum theory does not necessarily happen between paths in real position space, but in general happens between paths in abstract configuration space. And those “paths” in configuration space can sometimes correspond to entire processes.

With the above knowledge, two aspects can be put forward. First, there is no photon origin problem for one-photon interference in a three-alternative MZI because there is no photon generation process. Note that the path information we are discussing in this work refers to the trajectory along which we can trace back to where the photon was born. With one-photon interference in a three-alternative MZI, a photon originates only from the input port of the MZI; therefore, we know its source.

Again, I don't quite agree with this reasoning. Whether there is a photon origin question or a photon trajectory question is a matter of detail, but as far as quantum theory is concerned, both of these correspond to a “which-path” question where the path is in configuration space.

Second, no grouping can be interpreted in the one-photon three-alternative MZI. Imagine three paths A, B, and C, one group the A and B together and tuning the relative phase between A and B to be π , can we say that the photon only passes through C, not through A and B? Definitely not. It's only when the photon also takes the paths through A and B that we can say there is destructive interference; otherwise, how can the destructive interference from A and B come from?

The authors had also made a similar comment previously as

“One cannot say this photon only passes through C and not through A and B; otherwise, how can one prove that A and B have a destructive interference if it only passes through C?”

I am not convinced by this reasoning. **Just because there is destructive interference does not mean the photon actually took those paths.**

For instance in the Hong-Ou-Mandel experiment, there are two alternatives in which the two photons come out from different ports that destructively interfere. The above reasoning would suggest that the two photons indeed come out of the two different ports otherwise how can destructive interference occur? This does not seem consistent to me.

“However, in a three nonlinear crystal interferometer, A and B crystals can be grouped together to interpret, as they have their own photon generation process. When these two processes from A and B have destructive interference, we can say that the photon originates from crystal C. This is definitely different from the one-photon three-alternative MZI.”

Again, I do not see how this is fundamentally different from the one-photon linear MZI case. If there is destructive interference between the two processes A and B, nothing can be said about what really happened there. For instance, one could claim that down-conversion happened in A followed by upconversion in B. Or alternatively, one could claim that the pump photon passed through both A and B completely unaffected. I am not sure if one can solidly claim one way or another.

Regarding the following explanation by the authors

In fact, this and the next comments also pose the fundamental question: What are the differences between the nonlinear interferometer (an interferometer composed of parametric amplifiers) invested here and the traditional linear interferometer (an interferometer composed of beam splitters), such as the Mach-Zehnder interferometer (MZI)? As already proposed and thoroughly analysed by [Yurke et al., Phys. Rev. A 33, 4033 (1986)], the nonlinear interferometer incorporates a $SU(1,1)$ transformation that describes the evolution under parametric amplification, while the traditional MZI is governed by the $SU(2)$ transformation, which is an evolution of two-mode linear optics. In addition, $SU(1,1)$ transformations describe active optical dynamics, which produce a non-classical Gaussian state of light inside the $SU(1,1)$ interferometer, even from vacuum input. From the perspective of state evolution, the photon number is conserved in a traditional MZI; however, the photon number is not conserved due to amplification in a nonlinear interferometer, and it also involves squeezing and entanglement generation. Other literature that compares the differences can also be found in [Hudelst et al., Nature Commun. 5, 3049 (2014); arXiv2505.15635 (2025)].

I thank the authors for this explanation, but these differences have no bearing on my observation.

Response: We thank the reviewer for the careful and insightful comments. We agree with the reviewer that, at a fundamental level, interference in quantum theory always arises from the linear superposition of probability amplitudes associated with alternative histories in an appropriate configuration space. In this abstract sense, interference between “path alternatives” and interference between “process alternatives” can indeed be regarded as manifestations of the same underlying quantum principle.

However, the purpose of the present work is not to distinguish different forms of interference at this most abstract level, but rather to **analyze how which-path (or which-source) statements acquire meaning in concrete experimental contexts**, and how different groupings of alternatives may or may not be operationally justified. The central point of our paper is therefore not about the formal superposition principle itself, but about the **physical and interpretational content** that can be ascribed to different alternatives in specific interferometric architectures. We believe this clarification adequately addresses the reviewer’s concern.

- **R1.3:** Regarding whether the word “subjective” is appropriate in the title of the paper, I understand the authors’ point of view now. In this sense, a lot about the underlying interpretations in quantum theory can be “subjective” and so I am fine with the authors sticking to that word. The word appears three times in the paper. The last time it appears is on line 131 as

“This result arises because the application of quantum mechanics to the actual system is inherently subjective”

I understand the context in which this sentence appears, but I would suggest the authors to rephrase this sentence a bit. There is really no ambiguity or subjectivity in the application of quantum mechanics as a formalism to the system under consideration. Rather there is a “subjectivity” in the story one would like to tell about what is really happening in the process. Besides, if the application of the formalism itself were subjective, then that could potentially threaten to cast quantum mechanics outside the purview of science.

Response: We thank the reviewer for this helpful comment and for their constructive view regarding the use of the term “subjective” in the title. We fully agree with the reviewer that **there is no subjectivity in applying quantum mechanics as a formal and predictive framework** to the physical system under consideration.

The intention of the sentence on line 131 was not to suggest any ambiguity or arbitrariness in the quantum-mechanical formalism itself, but rather to emphasize that **different, equally valid**

interpretational narratives can be constructed to describe the same underlying quantum evolution, depending on how one chooses to decompose or group the contributing probability amplitudes.

We agree that the original wording could be misinterpreted as implying that the formalism of quantum mechanics is itself subjective, which was not our intention. To avoid this ambiguity, we have rephrased the sentence in the revised manuscript to clearly distinguish between the objectivity of the formalism and the subjectivity of the interpretational description.

Specifically, the sentence

“This result arises because the application of quantum mechanics to the actual system is inherently subjective”

has been revised to

Line 131: **“This result arises because, while the quantum-mechanical formalism provides an unambiguous and objective description of the system, the interpretational narrative used to describe the underlying processes can be subjective.”**

We thank the reviewer for this suggestion, which has helped us significantly improve the clarity and scientific precision of the manuscript.

R1.4: In relation to the question about whether this phenomenon is unique to nonlinear interferometers or not, the authors have provided very detailed arguments to the effect that it is indeed unique, but no revisions have been made to the manuscript in this regard. I personally did not find those arguments entirely convincing, but if the authors do not intend to bring up the issue in the paper at all, then perhaps it is fine. However, if the authors wish to explicitly contend that the phenomenon is peculiar to nonlinear interferometers and has no analog in one-photon interference, then a detailed discussion is warranted in the paper.

In summary, I am now happy to recommend this paper for publication in Nature Communications.

Response: We thank the reviewer for this final comment and for their careful consideration of the manuscript. We appreciate the reviewer’s assessment regarding the question of whether the discussed phenomenon is unique to nonlinear interferometers.

We acknowledge that our arguments in the response letter concerning uniqueness may not be fully compelling at a fundamental level, and we agree with the reviewer that **a strong claim of strict uniqueness would require a much more extensive and dedicated discussion**. We therefore clarify that it is **not our intention to contend in the manuscript that the phenomenon has no analogue in one-photon interference**.

Accordingly, in the revised manuscript, we do not claim fundamental uniqueness. Instead, our discussion is limited to emphasizing that the phenomenon becomes **operationally and experimentally meaningful in nonlinear interferometers**, where distinct photon-pair generation processes can be physically enabled or disabled. Any broader statements about the absence of analogues in linear interferometers are intentionally avoided.

We believe that this positioning is consistent with the scope of the present work and avoids overstating the conclusions. We thank the reviewer for highlighting this point and for their positive overall recommendation.

Changes in the revised manuscript:

Line 461: "Another interesting point is to realise the subtle difference in the **operational and physical interpretation** of the "which-path information" depending on what measurement is performed."

Line 539: "This brings about a refinement of the **physical interpretation and experimental description** of "which-path information"."

Reviewer #2 (Remarks to the Author):

R2: The paper can be published now. I'm happy to have alerted the authors to the apparently open problem of developing entropic uncertainty relations which satisfactorily describe the tradeoff between visibility and an appropriate notion of distinguishability for compound nonlinear interferometers.

Response: We thank the reviewer for this positive assessment and for highlighting this important open problem. We agree that developing entropic uncertainty relations capable of capturing the tradeoff between interference visibility and an appropriate notion of distinguishability in compound nonlinear interferometers remains an interesting and largely unexplored direction.

While such an analysis is beyond the scope of the present work, we believe that our results provide a useful physical setting and motivation for future investigations along these lines. We appreciate the reviewer's insightful remark and are grateful for bringing attention to this broader perspective.

Reviewer #4 (co-reviewed with Reviewer #3, Remarks to the Author):

R4: We acknowledge the great effort by the authors in responding to the comments in our previous report.

Previously, the main point of the paper, and hence its value, was kind of fuzzy and not clear to me. In particular, I think the modification in the abstract, as well as elaboration in the conclusion, makes it now clearer. To my understanding, the main point of this paper is to bring up (or in some sense, to remind, because I think actually this is not completely unknown to the physicists, but usually we don't really care about this because we don't worry about interpretation) the point that, as the authors say, "Any probability could, in principle, be subdivided into alternatives of non-zero probability amplitude, which can lead to a zero-probability amplitude of the combined probability. Therefore, "which-path information" can only be meaningfully defined if the context is clarified and the alternatives are explicitly specified in this context" (The earlier version using "event" was somehow making it confusing). In fact, I think this sentence in the conclusion perhaps should be put earlier in the introduction, as this is really the central message in my opinion (which in fact goes beyond the discussion about nature of path itself, which now serves as a specific demonstration).

For the other points that the authors replied, and all the additions in the Supplementary material, as well in the main text, and corrections of typos, we are mostly OK with that.

Now, as for whether the work is suitable for publication in Nature Communications, to be honest, we remain somewhat reserved. As we said in our earlier report, we value that "this work is thought-provoking and can potentially help us appreciate better", now instead of "the role of interference in a "which-source" setup", I would say "the identification of 'alternatives' versus which-path information", still, I'm not certain if this work really brings something that is really fundamental for new physics in the future. So, I'm not completely against its publication in Nat. Commun., but I'm not completely recommending it either.

Response: We sincerely thank the reviewer for their careful re-evaluation of the revised manuscript and for the thoughtful and fair assessment provided in this report.

We are particularly grateful that the reviewer has identified and articulated what they consider to be the central message of our work. We fully agree with their understanding that the main point of this paper is to emphasize that *which-path information is meaningful only once the relevant set of alternatives and the experimental context are explicitly specified*. We appreciate the reviewer's remark that this point may not be entirely unknown, but is often left implicit or overlooked because interpretational issues are usually not the focus.

We also agree with the reviewer's valuable suggestion that the sentence currently appearing in the conclusion—

“Any probability could, in principle, be subdivided into alternatives of non-zero probability amplitude, which can lead to a zero-probability amplitude of the combined probability. Therefore, ‘which-path information’ can only be meaningfully defined if the context is clarified and the alternatives are explicitly specified in this context.”

—captures the core message of the paper particularly well. Following this suggestion, we have moved a revised version of this statement to the Introduction, where it now clearly frames the scope and motivation of the work at an early stage. The subsequent discussion of path, source, and process alternatives is then presented explicitly as a concrete and experimentally relevant demonstration of this general principle.

Regarding the reviewer's remaining reservation about the suitability of the work for Nature Communications, we fully respect this perspective. Our intention is to provide a **conceptually clarifying contribution** that is broadly relevant across quantum optics, quantum information, and interferometry. We believe that explicitly identifying the role of context-dependent alternatives in defining which-path information is timely and valuable, particularly given the increasing use of complex interferometric architectures—linear and nonlinear alike—where such distinctions are often invoked but rarely made precise.

We hope that the revisions made, especially the clearer framing of the main message in the Introduction and Conclusion, help convey that the value of this work lies in sharpening conceptual understanding and avoiding potential misinterpretations, rather than in proposing new physics per se.

We thank the reviewer again for their constructive feedback, which has significantly improved the clarity, focus, and presentation of the manuscript.

Changes in the revised manuscript:

Line 142: “In general, a given probability can be decomposed into alternative probability amplitudes, each of which may be non-zero, while their coherent superposition leads to a vanishing total amplitude. This implies that “which-path information” can only be meaningfully defined if the context is clarified and the relevant alternatives are explicitly specified in this context.”

Review Report

I have carefully read the manuscript titled “*Subjective nature of path information in quantum mechanics*”. In this work, the authors describe a quantum optical interference experiment in which the which-path question appears to admit two distinct but equally self-consistent answers. In what follows, I outline my queries and comments regarding this work:

- There appears to be a contradiction between

sentence (line 38) “*We demonstrate the simultaneous observation of zero interference visibility and the complete absence of which-path information using a three-crystal interference setup.*”

and

sentence (line 123) “*Unlike the two-crystal case, ascribing a definite origin to a photon pair is impossible even though full “path information” is available and no interference is observed.*”

The authors may please correct this discrepancy.

- On line 72, the word “outcomes” may be replaced by “path histories”. Also, “unpredictable” does not seem fully accurate because this is a statement about retrodiction and not prediction. Perhaps “unascertainable” could be a better choice.
- On line 100, the sentence reads “*Frustrated down-conversion thus provides a platform to study the interplay between indistinguishability, interference, and which-source information.*” Perhaps the word “interference” may be redundant here because the tradeoff is between indistinguishability and which-path information.
- On line 118, the sentence reads “*This stems from the different possibilities of partitioning reality into events that are represented by probability amplitudes.*” As the term “event” refers to something that happens at a given location in space and instant in time, it does not seem appropriate here. I think the word “events” may be replaced by “path histories” for accuracy.
- On line 223, “*Now, we can apply the duality relation between the observed visibility*” may be rephrased to “*Now, one can apply the duality relation between the observed visibility*”. Similarly, in line 278 “*Thus, we can conclude that the photons were emitted by S1’ , that is, NL1.*” may be rephrased to “*Thus, one can conclude that the photons were emitted by S1’ , that is, NL1.*”
- On line 317, the sentence reads “*Phase plates are placed between each pair of crystals and are used to tune the relative phase (ϕ_A and ϕ_C) between the pump, signal, and idler photons.*” In my view, the phrase “*between the pump, signal, and idler photons*” is vague. The quantities ϕ_A and ϕ_C are relative phases between probability amplitudes corresponding to the path alternatives of the process from the source to the detector.
- In the caption for Fig 3, there are multiple typos. Please correct them.

- On line 455, the sentence reads “In our experiment, the visibilities of the interference between S1 and S2 and between S1' and S2' when setting $\phi_A = \pi$ and $\phi_C = \pi$ are $9.12 \pm 3.81\%$ and $8.30 \pm 4.19\%$, respectively.” Here S1, S2, S1', S2' refer to effective sources. Strictly speaking, the interference is not between sources but rather between probability amplitudes corresponding to downconversion happening at those sources. Although I agree that brevity and simplicity may warrant some flexibility, the authors may want to explore other rephrasings. Also, do these visibilities follow any duality relation or tradeoff with respect to the path probabilities p_3 and p_1 calculated on line 492-294? If yes, I would suggest that the authors point this out.
- On line 493, the sentence in the parentheses “(the probability is calculated by $CCNL1 / CC_{tot}$ and $CCNL3 / CC_{tot}$, where $CCNL1 = CC_{tot} - CCS2'$ and $CCNL3 = CC_{tot} - CCS1$, $CCS1 / CCS2'$ are counts when crystal NL3/NL1 is off and the phases are set to π , CC_{tot} are the counts when all three crystals are on and the phases are set to π)” is confusing. It is not clear which quantity refers to p_1 and which refers to p_3 . Please specify explicitly. Also, when the authors say “the phases are set to π ”, they are probably referring to ϕ_A and ϕ_C , but it is better to specify as “both phases ϕ_A and ϕ_C are set to π ”.
- On line 524, the authors write “However, the assignment of a probability amplitude of zero to an event should not be interpreted as a zero probability that this event will occur.” As I have pointed out before, the use of the word “event” is problematic. The authors are referring to an entire “path history” or “path alternative” from the pump source to the detector. The probability amplitudes correspond to these alternatives, and not to specific events.
- On line 526, the authors write “In the experiment of frustrated down-conversion [8], ideally no photon pairs are detected if there is complete destructive interference between two photon-pair emission processes. Nevertheless, the possibility for a photon pair originating from frustrated down-conversion cannot be ruled out if a third nonlinear crystal were inserted before the detectors. Therefore, a measured probability of zero for an isolated subsystem does not mean that this probability remains zero in the presence of other subsystems [19], since all subsystems can interfere with each other.”

I disagree with the above reasoning. When an additional crystal is added to the system, the system has changed in a non-trivial manner. There is no reason to expect that the photon pair rate would remain zero. Mathematically speaking, the total probability amplitude for the entire process must now an additional term. Contrary to what the authors say, the sum of the two original terms still remains zero. It is the introduction of the new term corresponding to the third crystal that will now make the total photon pair rate non-zero. There is nothing confusing or surprising about this. The situations with and without the third crystal are two completely different situations.

Also, it is not accurate to say “subsystems can interfere with each other” because interference does not happen between systems (or subsystems) per say. This issue was at the heart of Dirac’s remark that a photon only interferes with itself, and not with any other photon. Roy Glauber, for instance, later clarified that interference happens between probability amplitudes corresponding to path histories or path alternatives. And these alternatives are specified all the way from the original source to the detection stage. This is the basic premise of even the path integral formalism pioneered by Feynman.

- On line 561, the authors write “The one-particle and two-particle interference are widely characterized in the framework of the duality relation. This interference phenomenon has its roots in the fundamental indistinguishability of identical quantum particles.”

I disagree with this remark. The origin of interference is not the indistinguishability or the identicalness of the particles themselves, but rather of the alternatives or the path histories of the system as a whole. That system could be a one-particle system or a two-particle system depending on the kind of experiment being performed.

Note that if what the authors state was true, it should never be possible to observe two-photon interference effects with non-degenerate entangled photons from parametric down-conversion. However, Hong-Ou-Mandel-type effects and other two-photon interference effects have been observed with two-photon systems comprising of non-identical photons. In fact, the famous postponed compensation experiment reported in the paper titled “Can two-photon interference be considered the interference of two photons?” *Phys. Rev. Lett.* **77**, 1917 (1996) shows a Hong-Ou-Mandel-type effect with the photons not even appearing at the beam-splitter at the same time. Such experiments clearly demonstrate that it is not the indistinguishability or identicalness of the photons themselves that counts, but rather of the alternatives of the evolution of the system as a whole.

- On line 573, the authors write “Our experiment also demonstrates one important feature of quantum mechanics: the description of the quantum system must encompass all the possible outcomes, as long as no intervention is made to make the parts distinguishable in principle. Whether the quantum system should be treated as a whole or as separable parts depends on whether the parts can be distinguished from each other.”

I don’t quite agree with this claim. Strictly speaking, the experiment has only two possible outcomes, namely, either the photon pair is detected or it is not. But the description must encompass all the possible alternatives or path histories, not outcomes. In addition, it is not really a question of making the parts distinguishable, but rather about making the alternatives distinguishable. The quantum system under investigation is the two-photon field, not the crystals. And it is imperative to treat this two-photon system as a whole right from the pump source to the detectors. The present study does not raise any questions about this matter.

- On line 586, the authors write “By considering two of the three sources as a single source, two contradictory interpretations of the path information are possible within the experimental configuration.” I am not sure “contradictory” is the right word here. Within the quantum formalism, there is no really no contradiction. There are two distinct common-sense interpretations that are equally self-consistent but mutually incompatible from a common-sense point of view.
- On line 591, the authors write “Usually, we identify macroscopically distinguishable “events” and assign probability amplitudes to them. However, this identification of “events” is generally ambiguous. Any event could in principle be subdivided into subevents of non-zero probability amplitude, which can lead to a zero probability amplitude of the combined event. Therefore, the “which-way information” can only be meaningfully defined if the context is clarified and the events are explicitly specified in this context.”

Again, I disagree with the choice of the word “events” as I have indicated throughout my

review. The authors are referring to “path histories” that are further divided into “sub-histories”. This division, when applied to three-alternative interference experiments like the one described here, admits distinct interpretations about “which-path information” depending on how the division is applied.

- On line 796 in the Methods section, the authors write “After recursive optimisation, the observed visibility between NL1 and NL2 is $98.53 \pm 0.18\%$ and the visibility between NL2 and NL3 is $98.68 \pm 0.17\%$. This indicates that each pair of crystals exhibits a high degree of coherence.” Is there any estimate about what this visibility might be for the third pair, namely, NL1 and NL3? After all, the experiment is performed when all the three crystals are active. If there is any assumption involved here, the authors must explicitly state it.
- I do not find the title of the paper “*Subjective nature of path information in quantum mechanics*” to be adequately justified. In my view, this work points to an inherent ambiguity in the interpretation of path information when applied to three-alternative interference experiments. I don’t see this to be a question of subjectivity versus objectivity. If it was subjective, then I would expect that each experimenter would arrive at some definite conclusion about the path information, but different experimenters would be entitled to different conclusions. But that is not the case in this work. Any experimenter would face an ambiguity on the question of path information, and in that sense, it is really an objective ambiguity.
- I don’t think that this phenomenon is peculiar to nonlinear interference experiments of the kind described in this work. As such, the ambiguity in path information has nothing to do with two-photon entanglement. Therefore, I expect that this ambiguity would persist even in linear interference experiments that involve three or more alternatives. For instance, if one could consider one-photon interference in a three-alternative linear Mach-Zehnder interferometer, I would expect that this ambiguity would affect that setup as well. Can the authors comment on this matter?
- In fact, if I may take the previous observation one step further, doesn’t the essentially same phenomenon occur even in classical wave interference in a three-alternative Mach-Zehnder interferometer? If one were to set up the interferometer in such a way that the three paths have relative phases $\phi_A = \pi$ and $\phi_C = \pi$, then one could equally consistently interpret the output wave as resulting from a cancellation of wave amplitudes in arm 1 and arm 2, or alternatively in arm 2 and arm 3. I understand that this situation involves a superposition of classical wave amplitudes instead of quantum probability amplitudes, but there is essentially the same ambiguity about whether the energy of the output wave traversed arm-1 or arm-3. Could the authors comment on this as well?

Overall remarks:

I find this work to be quite interesting from a fundamental point of view. It has the potential to spark debates on the nature of path information in quantum interference. I find the manuscript to be very well-written and the experimental observations to be well analyzed and presented. However, I am still unclear about the precise implications of these observations. As I have explained in this review, I don’t agree with the interpretations that have been presented by the authors. As a result, I am unable to recommend the publication of this paper in its current form. However, if the authors carefully consider my comments and are able to address my concerns to my satisfaction, I would be willing to reconsider my decision in the next round of review.

Review Report for Round 2

I have carefully read the response letters submitted by the authors. I thank the authors for making such a comprehensive effort at addressing each concern/suggestion that I had raised in the previous round of review. In particular, I found many of their explanations quite insightful and thought-provoking.

The authors have resolved most of the concerns and suggestions that I had raised in the first round of review. In what follows, I would just like to outline a few residual concerns from my side. I mark the authors' comments from their response letter in blue normal font, *excerpt from the manuscript in blue italics*, and my own remarks in black normal font.

Firstly, I am still not convinced that the phenomenon reported in this work is peculiar to nonlinear interferometers and has no analog in linear one-photon interferometers.

- The authors state that “Note that the meaning of path information in our three-crystal interferometer and the three-path linear Mach-Zehnder interferometer (MZI) is very different. In the MZI, the path information refers to the path the photon passes through. In the three-crystal interferometer, the path information refers to the place where the photon is generated. In the one-photon three-path linear MZI, the one photon must pass through all three paths, even though one blocks C and then blocks A&B when trying to regard A&B as path S1 and C as another path S2 (see Fig.~3). Because there is only one photon, one-third of the probability that the photon passes through each path...One cannot say this photon only passes through C and not through A and B; otherwise, how can one prove that A and B have a destructive interference if it only passes through C? However, with the three-crystal case, one can conclude that the detected photon pairs originate from crystal C=NL3 because it is reasonable that the detected photons are only generated by C and travel to the detectors when A and B are blocked.”

I am not sure I agree with this reasoning. In both the three-crystal case and the linear MZI case, one can imagine a destructive interference between A and B. Alternatively, one can also imagine a destructive interference between B and C.

In this regard, I think it is important to not get confused by the parallel geometry of the linear SU(2) Mach-Zehnder interferometer and the series geometry of the nonlinear SU(1,1) interferometer. Although the setup geometries are different, as far as the superposition of probability amplitudes corresponding to different alternatives is concerned, both situations are perfectly analogous.

- The authors then state that “Thus, we can interpret the traditional MZI in two dual ways: (1) Path interference (path histories or path alternatives, as often mentioned by the Reviewer): A photon can take two paths, and the outputs result from the quantum superposition of these paths, which is a particle picture. (2) Field interference (as mentioned by the Reviewer's comment R1.19): Optical fields split and recombine, and their amplitudes interfere, which is a wave picture. However, in the nonlinear interferometer, the interference occurs between “the vacuum evolving through the second crystal” and “the amplified field coming from the first crystal” [Yurke et al., Phys. Rev.. A 33, 4033 (1986)]. In this sense, it's not just an interference of different path alternatives, but an interference between two processes, i.e., coherent superpositions of different pair-generation amplitudes. Therefore, we can refer to it as the interference between probability amplitudes corresponding to process alternatives.”

I would just still argue that on a fundamental level, there is no difference between interference of different path alternatives and the interference of difference process alternatives. This is because **the notion of “paths” in quantum theory** does not necessarily refer to paths in real position space, but **in general refers to “paths” in an abstract configuration space**. In the linear MZI, these paths just happen to correspond to actual paths that a photon can take in real space, whereas in the 3-crystal nonlinear interferometer, these “paths” correspond to ways in which the pump photon can get destroyed to produce a two-photon state. But in the end, it is still the same principle of linear superposition in action and results in interference between amplitudes corresponding to these paths.

Also, regarding the statement that in a nonlinear interferometer “the interference occurs between “the vacuum evolving through the second crystal” and “the amplified field coming from the first crystal”[Yurke et al., Phys. Rev. A 33, 4033 (1986)], I am not denying this interpretation but one could equally well argue that the interference occurs between probability amplitudes corresponding to the different alternatives by which down-conversion can occur. For instance, please see Appendix A of the paper P. Sharapova et al. Phys. Rev. A **91**, 043816 (2015). The equation A1 has been obtained by linearly superposing the probability amplitudes for down-conversion happening in crystal 1 and crystal 2.

I also looked at the reference cited here:

“This process interference was even demonstrated in the experiment, when the two processes are distantly separated [Pseiner et al., Phys. Rev. Res. 6, 013294 (2024)]. Note that this is not two-photon interference, despite having two photons.”

I do not see how this refutes my observation. However, if the authors have only cited this reference to argue that this is interference between processes and not paths in real space, then I agree with them. But as I have pointed out previously, interference in quantum theory does not necessarily happen between paths in real position space, but in general happens between paths in abstract configuration space. And those “paths” in configuration space can sometimes correspond to entire processes.

With the above knowledge, two aspects can be put forward. First, there is no photon origin problem for one-photon interference in a three-alternative MZI because there is no photon generation process. Note that the path information we are discussing in this work refers to the trajectory along which we can trace back to where the photon was born. With one-photon interference in a three-alternative MZI, a photon originates only from the input port of the MZI; therefore, we know its source.

Again, I don't quite agree with this reasoning. Whether there is a photon origin question or a photon trajectory question is a matter of detail, but as far as quantum theory is concerned, both of these correspond to a “which-path” question where the path is in configuration space.

Second, no grouping can be interpreted in the one-photon three-alternative MZI. Image three paths A, B, and C, one group the A and B together and tuning the relative phase between A and B to be π , can we say that the photon only passes through C, not through A and B? Definitely not. It's only when the photon also takes the paths through A and B that we can say there is destructive interference; otherwise, how can the destructive interference from A and B come from?

The authors had also made a similar comment previously as

“One cannot say this photon only passes through C and not through A and B; otherwise, how can one prove that A and B have a destructive interference if it only passes through C?”

I am not convinced by this reasoning. **Just because there is destructive interference does not mean the photon actually took those paths.**

For instance in the Hong-Ou-Mandel experiment, there are two alternatives in which the two photons come out from different ports that destructively interfere. The above reasoning would suggest that the two photons indeed come out of the two different ports otherwise how can destructive interference occur? This does not seem consistent to me.

“However, in a three nonlinear crystal interferometer, A and B crystals can be grouped together to interpret, as they have their own photon generation process. When these two processes from A and B have destructive interference, we can say that the photon originates from crystal C. This is definitely different from the one-photon three-alternative MZI.”

Again, I do not see how this is fundamentally different from the one-photon linear MZI case. If there is destructive interference between the two processes A and B, nothing can be said about what really happened there. For instance, one could claim that down-conversion happened in A followed by upconversion in B. Or alternatively, one could claim that the pump photon passed through both A and B completely unaffected. I am not sure if one can solidly claim one way or another.

Regarding the following explanation by the authors

In fact, this and the next comments also pose the fundamental question: What are the differences between the nonlinear interferometer (an interferometer composed of parametric amplifiers) invested here and the traditional linear interferometer (an interferometer composed of beam splitters), such as the Mach-Zehnder interferometer (MZI)? As already proposed and thoroughly analysed by [Yurke et al., Phys. Rev. A 33, 4033 (1986)], the nonlinear interferometer incorporates a $SU(1,1)$ transformation that describes the evolution under parametric amplification, while the traditional MZI is governed by the $SU(2)$ transformation, which is an evolution of two-mode linear optics. In addition, $SU(1,1)$ transformations describe active optical dynamics, which produce a non-classical Gaussian state of light inside the $SU(1,1)$ interferometer, even from vacuum input. From the perspective of state evolution, the photon number is conserved in a traditional MZI; however, the photon number is not conserved due to amplification in a nonlinear interferometer, and it also involves squeezing and entanglement generation. Other literature that compares the differences can also be found in [Hudelst et al., Nature Commun. 5, 3049 (2014); arXiv2505.15635 (2025)].

I thank the authors for this explanation, but these differences have no bearing on my observation.

- Regarding whether the word “subjective” is appropriate in the title of the paper, I understand the authors’ point of view now. In this sense, a lot about the underlying interpretations in quantum theory can be “subjective” and so I am fine with the authors sticking to that word. The word appears three times in the paper. The last time it appears is on line 131 as

“This result arises because the application of quantum mechanics to the actual system is inherently subjective”

I understand the context in which this sentence appears, but I would suggest the authors to rephrase this sentence a bit. There is really no ambiguity or subjectivity in the application of quantum mechanics as a formalism to the system under consideration. Rather there is a “subjectivity” in the story one would like to tell about what is really happening in the process. Besides, if the application of the formalism itself were subjective, then that could potentially threaten to cast quantum mechanics outside the purview of science.

In relation to the question about whether this phenomenon is unique to nonlinear interferometers or not, the authors have provided very detailed arguments to the effect that it is indeed unique, but no revisions have been made to the manuscript in this regard. I personally did not find those arguments entirely convincing, but if the authors do not intend to bring up the issue in the paper at all, then perhaps it is fine. However, if the authors wish to explicitly contend that the phenomenon is peculiar to nonlinear interferometers and has no analog in one-photon interference, then a detailed discussion is warranted in the paper.

In summary, I am now happy to recommend this paper for publication in Nature Communications.